# Proteostasis collapse, a hallmark of aging, hinders the chaperone-Start network and arrests cells in G1

**David F Moreno[1], Kirsten Jenkins[2,7], Sandrine Morlot[3,4], Gilles Charvin[3,4], Attila Csikasz-Nagy[2,7,5]\*, Martí Aldea[1,6]\***

[1]Molecular Biology Institute of Barcelona (IBMB), CSIC, Barcelona, Spain; [2]Randall Division of Cell and Molecular Biophysics, King's College London, London, United Kingdom; [3]Institut de Génétique et de Biologie Moléculaire et Cellulaire, Strasbourg, France; [4]Université de Strasbourg, Illkirch, France; [5]Faculty of Information Technology and Bionics, Pázmány Péter Catholic University, Budapest, Hungary; [6]Department of Basic Sciences, Universitat Internacional de Catalunya, Sant Cugat del Vallès, Spain; [7]Institute of Mathematical and Molecular Biomedicine, King's College London, London, United Kingdom

**\*For correspondence:**
attila.csikasz-nagy@kcl.ac.uk (AC-N);
mambmc@ibmb.csic.es (MíA)

**Competing interests:** The authors declare that no competing interests exist.

**Abstract** Loss of proteostasis and cellular senescence are key hallmarks of aging, but direct cause-effect relationships are not well understood. We show that most yeast cells arrest in G1 before death with low nuclear levels of Cln3, a key G1 cyclin extremely sensitive to chaperone status. Chaperone availability is seriously compromised in aged cells, and the G1 arrest coincides with massive aggregation of a metastable chaperone-activity reporter. Moreover, G1-cyclin overexpression increases lifespan in a chaperone-dependent manner. As a key prediction of a model integrating autocatalytic protein aggregation and a minimal Start network, enforced protein aggregation causes a severe reduction in lifespan, an effect that is greatly alleviated by increased expression of specific chaperones or cyclin Cln3. Overall, our data show that proteostasis breakdown, by compromising chaperone activity and G1-cyclin function, causes an irreversible arrest in G1, configuring a molecular pathway postulating proteostasis decay as a key contributing effector of cell senescence.
DOI: https://doi.org/10.7554/eLife.48240.001

## Introduction

Like most other cell types, individual yeast cells display a finite lifespan as they undergo subsequent replication cycles and, due to their relative simplicity, have become a very fruitful model to study the causal interactions among the different hallmarks of cell aging. Since yeast daughter cells are rejuvenated during most of the mother cell lifespan, it is generally accepted that aging is the result of asymmetric segregation of factors such as extrachromosomal rDNA circles (ERCs), dysfunctional mitochondrial and vacuolar compartments, or resilient protein aggregates (*Denoth Lippuner et al., 2014*; *Janssens and Veenhoff, 2016*). Proteostasis deterioration is a universal hallmark of cellular aging (*Kaushik and Cuervo, 2015*; *Klaips et al., 2018*; *Labbadia and Morimoto, 2015*; *López-Otín et al., 2013*), and yeast cells have been the paradigm to study the mechanisms of asymmetric segregation of protein aggregates or deposits and their relevance in aging (*Hill et al., 2017*).

Molecular chaperones play key roles in proteostasis by folding nascent polypeptides, refolding misfolded proteins, and facilitating their degradation or accumulation in different types of aggregates and deposits if they cannot be properly recycled (*Hartl et al., 2011*). Yeast cells display an age-dependent protein deposit, termed APOD (*Saarikangas and Barral, 2016*), that appears early

in their replicative lifespan and is retained in the mother cell compartment in every division cycle (*Hill et al., 2017*). Several chaperones including Hsp104, Ssa1 and Ydj1 co-localize with the APOD (*Andersson et al., 2013*; *Hill et al., 2014*; *Saarikangas and Barral, 2015*), where they are thought to play a concerted role in disaggregation and recycling of deposited proteins. Regarding asymmetric segregation of protein aggregates during cell aging, farnesylated Ydj1 has been shown to be important for proper retention of the APOD at the ER in the mother cell compartment (*Saarikangas et al., 2017*). The functional relevance of chaperones at the crossroads of protein aggregation and replicative aging is supported by the fact that Hsp104 and Ydj1 are required for a normal replicative lifespan and, when overexpressed, Hsp104 restores proteasome activity in aging cells (*Andersson et al., 2013*) and suppresses lifespan defects of *sir2* mutants (*Erjavec et al., 2007*). Moreover, by counteracting protein aggregation, overexpression of metacaspase Mca1 extends the lifespan of yeast mother cells in a Hsp104- and Ydj1-dependent manner (*Hill et al., 2014*).

The interdivision time of yeast cells increases during the last cycles before death (*Fehrmann et al., 2013*; *Lee et al., 2012*; *Lindstrom and Gottschling, 2009*) and most aging cells accumulate in the unbudded period before death (*Delaney et al., 2013*; *McVey et al., 2001*), suggesting that aging-related processes interfere with the mechanisms that trigger Start to drive cells into the cell cycle. The Cln3 cyclin is a rate-limiting activator of Start that is maintained at low but nearly constant levels during G1 (*Tyers et al., 1993*). Nuclear accumulation of Cln3 is driven by a constitutive C-terminal nuclear-localization signal (NLS) (*Edgington and Futcher, 2001*; *Miller and Cross, 2001*), but involves the essential participation of Ssa1 (or paralog Ssa2) and Ydj1 chaperones (*Vergés et al., 2007*) and the segregase activity of Cdc48 to release the G1 cyclin from the ER (*Parisi et al., 2018*). In addition, Ssa1 and Ydj1 also affect Cln3 stability (*Truman et al., 2012*; *Yaglom et al., 1996*), and their availability modulates the execution of Start as a function of growth and stress (*Moreno et al., 2019*). Here we study the effects of proteostasis decline during aging on the availability of Ssa1 and Ydj1 chaperones and, hence, on G1 cyclin function, aiming to uncover the processes that restrain proliferation in aged cells.

## Results

### Aging cells arrest mostly in G1 with low nuclear levels of cyclin Cln3 after the last budding event

To analyze cell-cycle entry kinetics in the last generations prior to death, we first examined wild-type cells expressing Whi5-GFP (*Costanzo et al., 2004*) in a CLiC microfluidics device (*Figure 1A* and *Video 1*) that had been developed for high-throughput analysis of single mother cells during aging (*Fehrmann et al., 2013*; *Goulev et al., 2017*). As previously observed, the average interdivision time was rather constant during aging until the senescence-entry point (SEP) (*Fehrmann et al., 2013*), when it displayed an abrupt increase that was maintained for ca. 2–3 generations on average prior to cell death (*Figure 1B*). The SEP concurred with an increase in the length of both unbudded (G1) and budded (S-G2-M) phases of the cycle. However, as assessed by the localization of Whi5 in the nucleus to inhibit the G1/S regulon (*de Bruin et al., 2004*; *Costanzo et al., 2004*), the G1 period prior to Start (T1) of the last three cycles before death displayed the largest relative increase compared to young mother cells (*Figure 1C*). Accordingly, while only about 15% of young mother cells are found in T1 in asynchronous cultures, the percentage of cells dying in this G1 subperiod increased up to ca. 75% (*Figure 1D*). Finally, old cells selected with the mother-enrichment program (MEP) displayed a larger fraction in G1 compared to young mother cells (*Figure 1—figure supplement 1A,B*). These data point to the notion that the deleterious effects of aging on cell cycle progression are particularly severe in G1 and prior to Start.

Execution of Start is particularly sensitive to growth, and cells arrest in G1 when deprived of essential nutrients. Nonetheless, old cells grew in volume after the last budding event at same rate as in the previous cycle (*Figure 1—figure supplement 1C*) and, as a result of progressive lengthening of G1, their size rapidly increased during the last cycles before death (*Figure 1—figure supplement 1D*). Our results agree with recent precise measurements of cell volume in aging cells until death in a different microfluidics device (*Sarnoski et al., 2018*). Overall, these data would rule out possible indirect effects of growth impairment on cell cycle progression in aging cells. On the other hand, the coefficient of variation in volume at the last budding event was 41.2%, while young cells

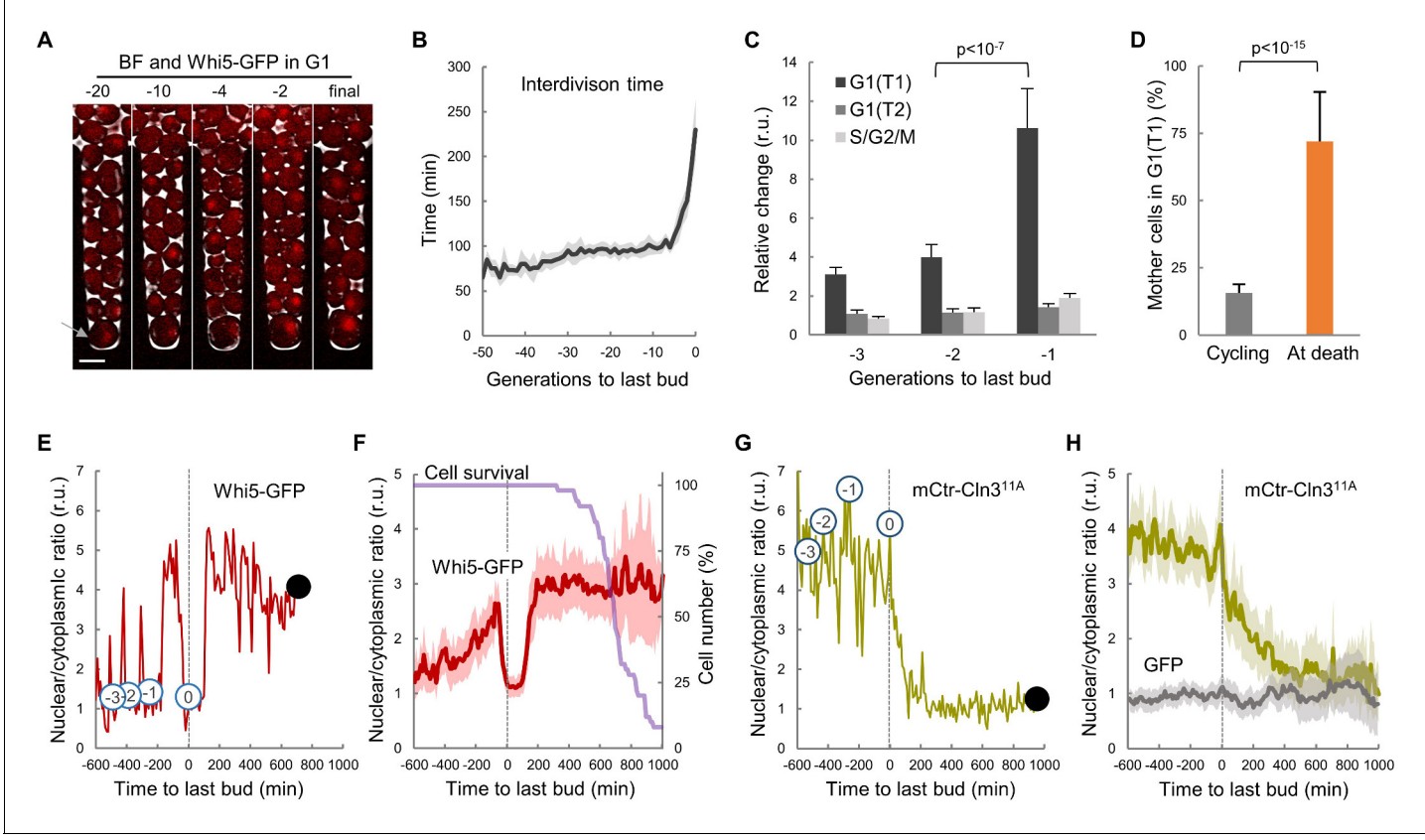

**Figure 1.** Yeast mother cells die mostly in G1 with low nuclear levels of cyclin Cln3. (**A**) A yeast mother cell (arrow) expressing Whi5-GFP during aging at the G1 phase of indicated cycles before death. (**B**) Interdivision times (mean ±CL, n = 50) aligned to the last budding event. (**C**) Cell-cycle period lengths (mean ±CL, n = 50) in aging cells relative to young cells. (**D**) Percentage (±CL, n = 50) of cells in G1 in young cycling cultures or at death. (**E**) Nuclear levels of Whi5-GFP during the last generations (numbers in open circles) before death (closed circle) of an aging mother cell. (**F**) Nuclear levels of Whi5-GFP (mean ±CL, n = 50) in aging cells as in panel E aligned at the last budding event. (**G–H**) Nuclear levels of mCtr-Cln3[11A] as in panels (**E**) and (**F**). Shown p-values were obtained using a Mann-Whitney U test. Bar = 5 µm. Results shown in this figure are representative of two replicate experiments.

DOI: https://doi.org/10.7554/eLife.48240.002

The following source data and figure supplement are available for figure 1:

**Source data 1.** Yeast mother cells die mostly in G1 with low nuclear levels of cyclin Cln3.
DOI: https://doi.org/10.7554/eLife.48240.004
**Figure supplement 1.** Yeast mother cells delay G1 progression with no major effects in growth during aging.
DOI: https://doi.org/10.7554/eLife.48240.003

displayed a reduced 20.2%, suggesting that cell size control at Start becomes less efficient as cells age.

To further characterize the observed defects in G1 progression in old mother cells, we carefully quantified the levels and localization of Whi5-GFP during the last cycles before death. In agreement with previous analyses of aging cells at the mRNA level (*Janssens et al., 2015*; *Yiu et al., 2008*), the overall cellular concentration of Whi5 did not change much during the last cycles (*Figure 1—figure supplement 1E*). The nuclear/cytoplasmic ratio oscillated normally in the previous cycles to the last budding event (*Figure 1E*), being high in G1 and low in the budded phases of the cycle due to Cdk-dependent phosphorylation and nuclear export of Whi5 (*de Bruin et al., 2004*; *Costanzo et al., 2004*). However, after the last budding event nuclear levels of Whi5-GFP remained low for about 200 min on average, and rose again to stay high in 76.8% of cells (*Figure 1F*), indicating that most cells completed the last cell cycle and arrested in the next G1 prior to death. Although the differences were not as large as previously reported (*Neurohr et al., 2018*), Whi5 levels in the nucleus displayed a 3-fold increase during the last cycles and the final arrest in G1.

Cln3 is the most upstream G1 cyclin acting in the positive feedback loop that inactivates Whi5 and executes Start (*Skotheim et al., 2008*; *Tyers et al., 1993*). Since Cln3 is too short-lived to be detected as a fluorescent-protein fusion in single cells, we used a hyperstable and hypoactive Cln3[11A] protein fused to mCitrine (mCtr-Cln3[11A]) that can be detected by fluorescence microscopy with no gross effects on cell cycle progression (*Schmoller et al., 2015*). As expected from its essential role in the nucleus, mCtr-Cln3[11A] displayed a distinct nuclear signal during the last cycles before the final budding event (*Figure 1G,H*, and *Video 2*); however, the nuclear/cytoplasmic ratio decreased to very low levels afterwards and remained low until death. In agreement with the fact that *CLN3* mRNA levels do not show significant changes in aged cells (*Janssens et al., 2015*; *Yiu et al., 2008*), overall cellular levels of mCtr-Cln3[11A] remained rather constant and similar to young cells (*Figure 1—figure supplement 1E*), ruling out major effects due to transcriptional or translational regulation of Cln3. In summary, our data suggest that aging cells would undergo profound alterations in the mechanisms that drive nuclear accumulation of cyclin Cln3 and, hence, delay G1 progression as observed in the last cycles before cell death.

## Ssa1/Ydj1 chaperone function is compromised in aging cells

We have previously shown that chaperones play a key role in the mechanisms that regulate Cln3 localization (*Moreno et al., 2019*; *Parisi et al., 2018*; *Vergés et al., 2007*). Ssa1 and Ydj1, with the participation of Cdc48, are important for releasing the G1 Cdk-cyclin complex from the ER and promoting its nuclear accumulation to trigger Start. On the other hand, it is generally assumed that aged cells display severe defects in protein homeostasis, thereby leading to the accumulation of misfolded-protein aggregates (*Kaushik and Cuervo, 2015*; *Klaips et al., 2018*; *Labbadia and Morimoto, 2015*). Thus, we decided to analyze the levels of Ssa1, Ydj1 and Hsp104 fused to fluorescent proteins during aging in the CLiC microfluidics chamber. Levels of Ssa1 and Ydj1 chaperones were only slightly reduced during the last cycles before death when compared to young cells (*Figure 2A*). By contrast, Hsp104 concentration rose steadily during aging until the last budding event (*Video 1*), when it reached a two-fold increase compared to young cells, and continued to increase afterwards during the posterior G1 arrest at an even higher average rate (*Figure 2A*). To confirm this result with a different experimental approach we used the mother enrichment program (MEP) (*Lindstrom and Gottschling, 2009*) to select cells

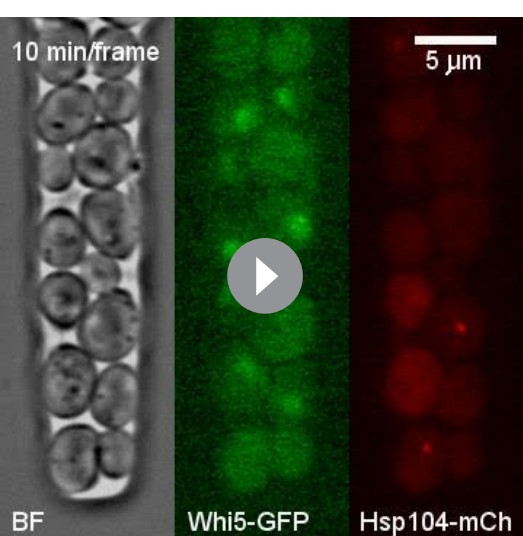

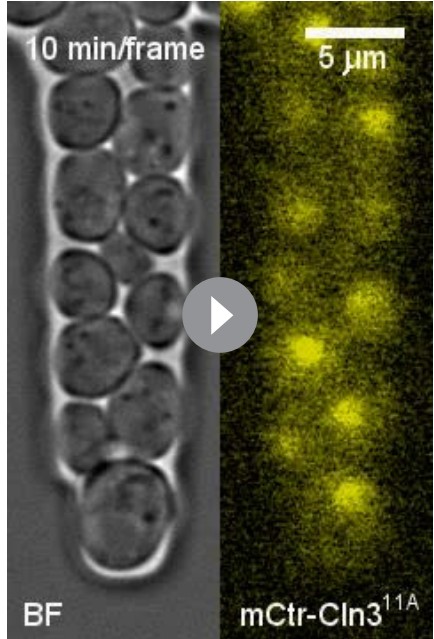

**Video 1.** Movie of a representative Whi5-GFP (green) Hsp104-mCh (red) cell in the CLiC microfluidic chamber. Images were taken every 10 min. The frame where the last budding event takes place is indicated.
DOI: https://doi.org/10.7554/eLife.48240.005

**Video 2.** Movie of a representative mCitrine-Cln3[11A] (yellow) cell in the CLiC microfluidic chamber. Images were taken every 10 min. The frame where the last budding event takes place is indicated.
DOI: https://doi.org/10.7554/eLife.48240.006

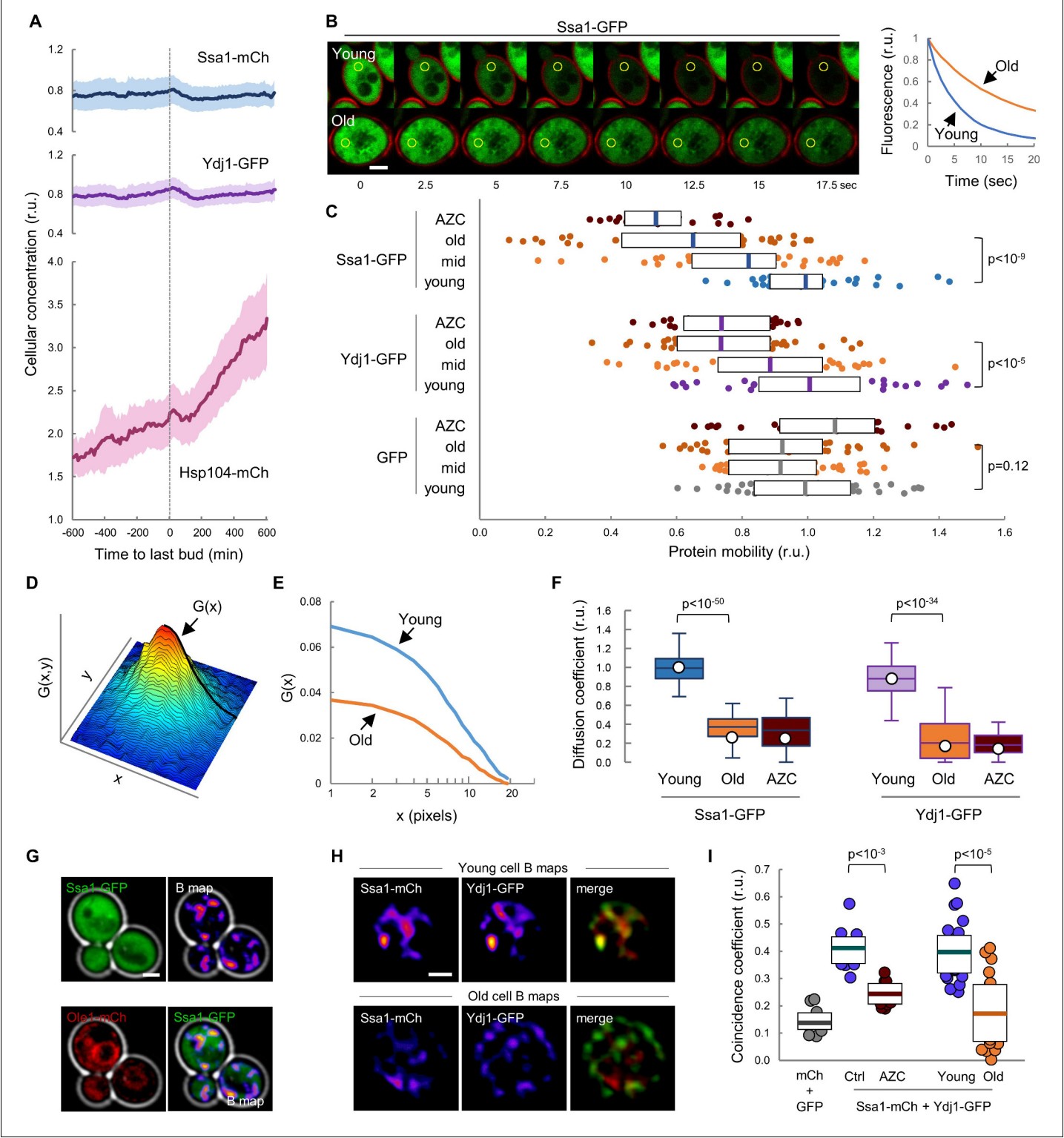

**Figure 2.** Mobility and spatio-temporal coincidence of Ssa1 and Ydj1 are reduced in aging cells. (**A**) Levels of Ssa1-mCh, Ydj1-GFP and Hsp104-mCh (mean ±CL, n = 50) in aging cells aligned at the last budding event. (**B**) FLIP analysis of Ssa1-GFP in representative young and old (MEP-aged) cells. (**C**) Mobility of Ssa1-GFP, Ydj1-GFP and GFP in MEP- cells aged for 24 hr (mid) or 48 hr (old), and young control or AZC-treated cells. Median ±Q (n = 40) values are also plotted. (**D**) Spatial autocorrelation function (ACF) by RICS for Ssa1-GFP. G(x), the ACF in the scanning direction, is indicated. (**E**) Average ACFs (n = 25) obtained by RICS for Ssa1-GFP in old (MEP-aged) and young cells. (**F**) Diffusion coefficients (open circles) obtained by RICS for Ssa1-GFP and Ydj1-GFP in old (MEP-aged) and young control or AZC-treated cells (n = 25). The results of Monte Carlo simulations (median ±Q, n = 66)

*Figure 2 continued on next page*

*Figure 2 continued*

are plotted. (**G**) Representative fluorescence intensity (top left) and brightness (B map, top right) images obtained by RICS for Ssa1-GFP. Ole1-mCh as ER reporter (bottom left) and merged (bottom right) images are also shown. (**H**) Representative brightness (**B**) maps obtained by RICS for Ssa1-mCh and Ydj1-GFP in old (MEP-aged) and young cells. Merged B maps with Ssa1-mCh (red) and Ydj1-GFP (green) are also shown. (**I**) Coincidence coefficients of Ssa1-mCh and Ydj1-GFP from B maps of old (MEP-aged) and young cells (n = 20), control and AZC-treated young cells (n = 10), and cells expressing GFP and mCh (n = 10). Median ±Q values are also plotted. Shown p-values were obtained using a Mann-Whitney U test. Bar = 2 μm. Results shown in this figure are representative of at least two replicate experiments.

DOI: https://doi.org/10.7554/eLife.48240.007

The following source data and figure supplements are available for figure 2:

**Source data 1.** Mobility and spatio-temporal coincidence of Ssa1 and Ydj1 are reduced in aging cells.

DOI: https://doi.org/10.7554/eLife.48240.010

**Figure supplement 1.** Chaperone availability is compromised in aging cells.

DOI: https://doi.org/10.7554/eLife.48240.008

**Figure supplement 2.** Correction of the mobility index obtained by FLIP as a function of cell size.

DOI: https://doi.org/10.7554/eLife.48240.009

aged for ca. 20 generations and also observed an increase in Hsp104 concentration (*Figure 2—figure supplement 1A*). Observed changes in chaperone concentrations agree with previous analysis at the mRNA (*Yiu et al., 2008*) and protein (*Janssens et al., 2015*) levels, and suggest that cells sense proteostasis defects and, regarding to Hsp104, react during aging similarly to other stress instances in which chaperone availability is assumed to be temporarily compromised (*de Nadal et al., 2011*). As their engagement in protein interactions must cause a decrease in the diffusion coefficient of chaperones, their mobility has been used as a proxy of availability (*Lajoie et al., 2012*; *Moreno et al., 2019*; *Saarikangas et al., 2017*). Thus, we used MEP-aged cells to analyze the mobility dynamics of Ssa1 and Ydj1 chaperones as GFP fusions by fluorescence-loss in photobleaching (FLIP). Notably, we detected a dramatic drop in mobility of both Ssa1 and Ydj1 when we compared aged cells with their young counterparts (*Figure 2B,C*). This decrease was similar to that caused in young cells by L-azetidine-2-carboxylic acid (AZC), which induces the accumulation of misfolded proteins with chaperones into disperse cellular aggregates (*Escusa-Toret et al., 2013*), thus compromising chaperone availability. By contrast, free GFP did not display significant changes in its mobility in aged or AZC-treated cells (*Figure 2C*). Since AZC treatment rapidly hindered nuclear localization of mCtr-Cln3[11A] (*Figure 2—figure supplement 1B*), these data point to the notion that aged cells would be impaired in their ability to accumulate Cln3 in the nucleus due to severe limitations in chaperone availability.

To further analyze chaperone mobility during cell aging we used Raster-Image Correlation Spectroscopy (RICS) (*Digman and Gratton, 2012*) as an orthogonal approach. Briefly, RICS provides information on moving molecules from raster-scan confocal images by obtaining an autocorrelation function (ACF) from small arrays of pixels within the cell (*Figure 2D*). After fitting a free-diffusion model to the autocorrelation functions of Ssa1-GFP from young and aged cells (*Figure 2E*), a significant drop in the coefficient of diffusion (D) of Ssa1-GFP was detected in aged cells (*Figure 2F*), which was again similar to that observed in AZC-treated young cells. Moreover, a similar behavior was observed for Ydj1-GFP (*Figure 2F*).

The intersection value of autocorrelation functions obtained by RICS depends on a second parameter related to the number of fluorescent molecules in the moving particles, termed brightness (B). Interestingly, aged cells displayed lower B values for both Ssa1-GFP and Ydj1-GFP compared to young cells (*Figure 2—figure supplement 1C*), which would reinforce the notion that the behavior of these two chaperones is altered in aged cells, perhaps as a result of different transient interaction dynamics. Contrary to the diffusion coefficient, which can only be robustly estimated after pooling data from many cells and images per cell, particle brightness can be determined rather consistently at a single-pixel resolution in every image to generate B maps. As shown in *Figure 2G*, Ssa1-GFP produced rather uneven B maps in young cells, displaying moving particles with more Ssa1-GFP molecules in compartments of the cell that did not particularly match the nucleus or the ER as assessed with an Ole1-mCh fusion (*Figure 2G*). We have recently described a procedure, called coincidence analysis (*Moreno and Aldea, 2019*), that uses B maps to study the spatio-temporal colocalization of molecular pairs undergoing transient interactions when performing their

function, such as Ssa1 and Ydj1. As previously observed, Ssa1-mCh and Ydj1-GFP displayed a much higher coincidence coefficient compared to free GFP and mCherry and, giving support to its application as a functional indicator of these two chaperones, their coincidence coefficient strongly decreased in the presence of AZC. Notably, B maps of Ssa1-mCh and Ydj1-GFP were more dissimilar in aged cells, and displayed a much lower coincidence coefficient compared to young cells (*Figure 2H,I*), suggesting that these two chaperones form less dynamic complexes in aged cells. All in all, these data point to the existence of important defects in the availability and concerted activity of Ssa1 and Ydj1, two key chaperones in the mechanisms that maintain protein homeostasis, in aged cells.

## Firefly luciferase aggregation takes place during the G1 arrest preceding cell death

Firefly luciferase (FFL) refolding and enzymatic activity recovery has been widely used to assay chaperone activity in vitro (*Glover and Lindquist, 1998*; *Schumacher et al., 1996*) and in vivo (*Nollen et al., 1999*), and an FFL-GFP fusion has been used as a single-cell reporter of chaperone activity after protein denaturation by heat shock (*Abrams and Morano, 2013*). We first compared the aggregation state of FFL-GFP in young and MEP-aged cells and found that, while we were unable to detect clear FFL-GFP foci in young cells, ca. 40% of cells aged for 20–25 generations showed a variable number of FFL-GFP foci (*Figure 3A,B*), confirming the notion that aged cells accumulate misfolded-protein aggregates. We then analyzed the dynamics of FFL-GFP aggregation during aging in the CLiC microfluidics chamber, and developed the required algorithms in BudJ (*Ferrezuelo et al., 2012*) to delimit and quantify fluorescent-protein aggregates with precision (*Figure 3C*). We detected the first visible FFL-GFP foci around the last budding event, followed by an accelerated increase in the amount of FFL-GFP present in foci until death (*Figure 3D,E*). It is important to note that, while Hsp104-mCh colocalized with FFL-GFP foci induced by heat shock in young cells as expected, most FFL-GFP foci in aged cells did not colocalize with Hsp104-mCh in the APOD (*Figure 3—figure supplement 1*), indicating that the normally operating mechanisms of misfolded protein recycling are altered in advanced aging.

As previously mentioned, overall Hsp104-mCh levels increased much faster after the last budding event (*Figure 3F*). However, Hsp104-mCh levels in the APOD remained constant, leading to a reduction of the Hsp104-mCh fraction in the APOD relative to total levels. Notably, the fraction of Hsp104-mCh in the APOD correlated at a single-cell level with the appearance of FFL-GFP foci in the following 180 min (*Figure 3G*), indicating that Hsp104 would not be able to accumulate in the APOD much before FFL-GFP aggregates are clearly visible. Since Hsp104 levels increase under stress conditions known to affect protein folding, our data reinforce the notion of proteostasis defects becoming increasingly important after the last budding event.

## Asymmetric aggregate inheritance predicts a decrease in chaperone availability and a G1 arrest in aging cells

The asymmetric distribution of protein aggregates to the mother cell during cytokinesis is a key safeguard mechanism to produce rejuvenated daughter cells (*Hill et al., 2017*). Thus, we established a stochastic model based on the asymmetric distribution of protein aggregates that appear stochastically during consecutive cycles of division, taking into account that chaperones are key factors in two mechanistic modules: (1) counteracting protein aggregation reactions and (2) facilitating nuclear accumulation of cyclin Cln3 to phosphorylate Whi5 and trigger Start (*Figure 4A*). Since Cln2 interacts with Ssa1,2 chaperones (*Gong et al., 2009*) and likely requires chaperoning activities similar to Cln3 (*Ferrezuelo et al., 2012*; *Moreno et al., 2019*), the model used a simplified version of Start without the positive feedback loop, and made Start strictly dependent on Cln3. Model structure (*Figure 4—figure supplement 1*), reactions (*Supplementary file 1*) and parameters (*Supplementary file 2*) were based on previous work by us (*Moreno et al., 2019*) as described in the Materials and methods section, and adjusted to obtain the replicative lifespan of wild-type cells, that is 30 cycles on average. First, we ran the model to simulate independent single cells, and stored all variables during consecutive cycles until a permanent G1 arrest was achieved, or up to a maximum time equivalent to 75 generations in wild-type cells under regular growth conditions. As shown in *Figure 4B*, simulated protein aggregates increased around the last budding event, causing a

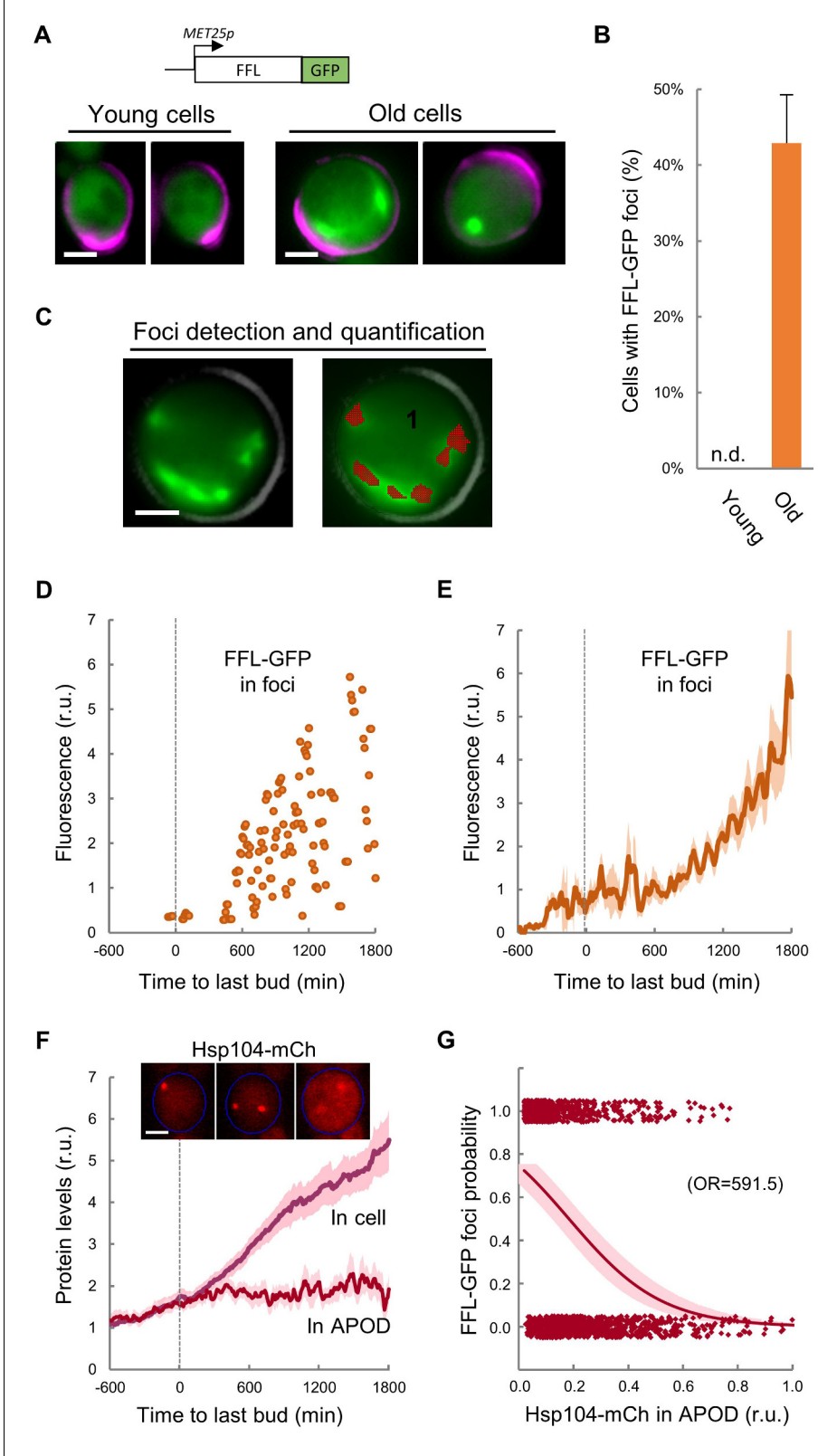

**Figure 3.** Firefly luciferase aggregates become visible during the last cycle before cell death. (A) Representative images of FFL-GFP expressed from a regulatable promoter in young and old (MEP-aged) cells. (B) Percentage (±CL, n = 230) of young and old (MEP-aged) cells with FFL-GFP foci. (C) Representative image of an old (MEP-aged) cell expressing FFL-GFP (left) with the foci overlay (red) obtained from BudJ. (D–E) FFL-GFP levels in foci (n = 50) aligned at the last budding event. Individual (D) and mean ±CL (E) data are plotted. (F) Cellular and APOD Hsp104-mCh levels (mean ±CL,

*Figure 3 continued on next page*

*Figure 3 continued*

n = 50) in aging cells aligned at the last budding event. A representative aging cell (inset) at 0, 600 and 1200 min after the last budding event is shown. (G) Probability of FFL-GFP foci within the following 180 min (twice the generation time of young mother cells) after reading the relative Hsp104-mCh levels in the APOD. Sampled single-cell data (closed circles) and the logistic regression line (mean ±CL) are plotted, and the obtained odds ratio is indicated. Bar = 2 µm. Results shown in this figure are representative of two replicate experiments.

DOI: https://doi.org/10.7554/eLife.48240.011

The following source data and figure supplement are available for figure 3:

**Source data 1.** Firefly luciferase aggregates become visible during the last cycle before cell death.

DOI: https://doi.org/10.7554/eLife.48240.013

**Figure supplement 1.** FFL-GFP and Hsp104 co-localization in heat-shocked and aging cells.

DOI: https://doi.org/10.7554/eLife.48240.012

sharp decrease in available chaperones (*Figure 4C*) and free nuclear Cln3 (*Figure 4D*). Notably, all these simulated variables displayed kinetics qualitatively similar to the experimental data (*Figure 4B, E insets*). Simulated interdivision time in consecutive cycles remained rather constant, but progressively increased during the last generations before the final G1 arrest (*Figure 4E* and *Figure 4—figure supplement 2A*), thus recapitulating the SEP (*Fehrmann et al., 2013*). Interestingly, the time when simulated levels of protein aggregates, available chaperones and free nuclear Cln3 initiated their respective changes closely correlated with the SEP (*Figure 4—figure supplement 2B–D*). Particularly for free nuclear Cln3 levels, which could be more precisely measured during the last division cycles before death, we observed a similar decrease to that predicted by the integrative model before and after the SEP (*Figure 4—figure supplement 2E*).

Next we perturbed the parameters of the model (*Supplementary file 3*) to qualitatively predict the effects of genetic ablation of the relevant factors or modification of important conditions such as growth rate and cell size. The *cln3* mutant exhibited a shorter lifespan (*Figure 4—figure supplement 3A*) as described (*Hill et al., 2014*; *Yang et al., 2011*), while the *whi5* knockdown mutant showed the opposite behavior and lived longer than wild-type (*Yang et al., 2011*). We also tested the effect of high and low growth rates in the model to simulate fast- and slow-growing cells. As experimentally observed (*Kaeberlein et al., 2005*; *Yang et al., 2011*), lifespan was strongly reduced by high growth rates (*Figure 4—figure supplement 3B*). Finally, since cell size has been proposed as a key factor affecting lifespan (*Yang et al., 2011*), we performed independent simulations of cells with different initial cell volumes and obtained a clear dependence of lifespan on initial cell size (*Figure 4—figure supplement 3C*).

## Cln3 overexpression increases replicative lifespan in a chaperone-dependent manner

The model predicted that increased levels of Cln3 would extend lifespan (*Figure 5A* inset). In order to measure the replicative lifespan of very large numbers of yeast cells we induced the MEP in cells growing in plates at low density, and obtained microcolonies with varying sizes that depended on the number of G2-arrested daughter cells produced by the mother cell during its replicative lifespan (*Figure 5—figure supplement 1A–C*). We first tested this experimental approach with wild-type and *cln3* cells (*Figure 5—figure supplement 1D*) and, as observed by conventional procedures (*Hill et al., 2014*; *Yang et al., 2011*), we found that Cln3 loss caused a ca. 40% reduction in lifespan. Next we used this approach to estimate the lifespan of cells overexpressing *CLN3* from a regulatable promoter and observed a remarkable increase in the relative lifespan compared to wild-type cells as predicted by the model (*Figure 5A*). Daughter cells overexpressing *CLN3* execute Start prematurely and bud at a smaller cell size (*Figure 5B*), which has been shown to have an effect on lifespan (*Yang et al., 2011*). To avoid these effects, we activated *CLN3* expression at different times after MEP induction, and compared the effects of *CLN3* overexpression in young cells and cells pre-aged for 24 hr (12–15 generations) and 48 hr (25–30 generations), respectively. Overexpressing *CLN3* in pre-aged cells did not affect their budding size (*Figure 5B*), but produced a similar relative increase in lifespan (*Figure 5C*). Thus, higher levels of Cln3 were able to increase lifespan independently of cell size.

While Ydj1-deficient cells displayed a reduced lifespan as previously observed (*Hill et al., 2014*), concurrent overexpression of Ssa1 and Ydj1 did not increase lifespan significantly (*Figure 5D*). Albeit

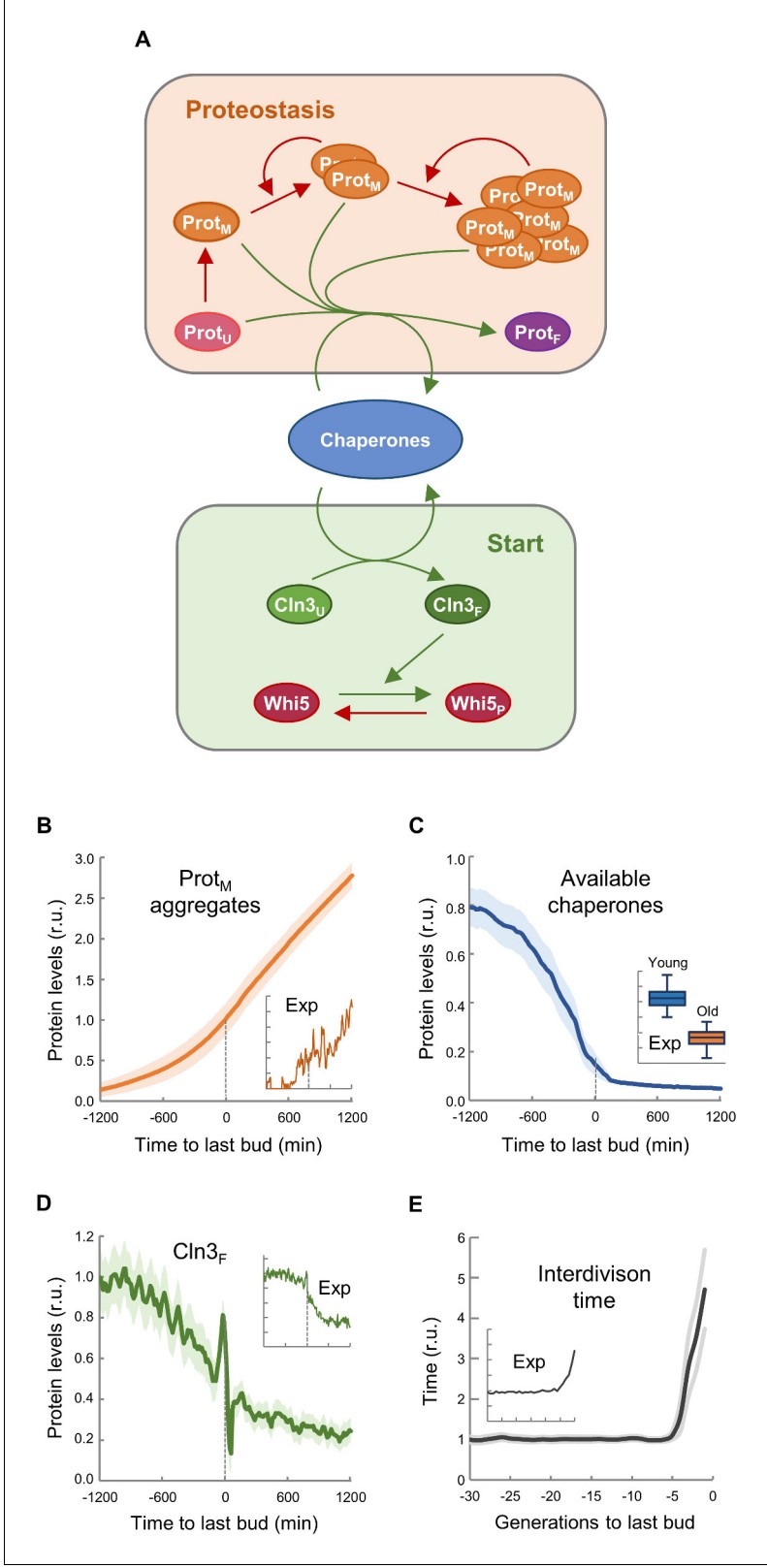

**Figure 4.** Asymmetric aggregate inheritance predicts a decrease in chaperone availability and a G1 arrest in aging cells. (A) Scheme of the integrative mathematical model with chaperones playing concurrent roles in proteostasis and Start. (B–E) Predicted aggregate protein (B), available chaperone (C), and free folded Cln3 (D) levels and interdivision times in aging cells aligned at the last budding event. Values (mean ±CL, n = 75) are plotted as lines. Experimental (Exp) data from *Figures 1B, H*, *2F* and *3E* are also shown as insets for direct comparison.

*Figure 4 continued on next page*

*Figure 4 continued*

DOI: https://doi.org/10.7554/eLife.48240.014

The following source data and figure supplements are available for figure 4:

**Source data 1.** Asymmetric aggregate inheritance predicts a decrease in chaperone availability and a G1 arrest in aging cells.
DOI: https://doi.org/10.7554/eLife.48240.018
**Figure supplement 1.** Wiring diagram of the integrative mathematical model.
DOI: https://doi.org/10.7554/eLife.48240.015
**Figure supplement 2.** The integrative model predicts a decrease in chaperone availability and free Cln3 at the first generation after the SEP.
DOI: https://doi.org/10.7554/eLife.48240.016
**Figure supplement 3.** Lifespan of mutants and growth conditions as predicted by the integrative mathematical model.
DOI: https://doi.org/10.7554/eLife.48240.017

surprisingly, our model predicted that chaperone overexpression would have a very limited effect on lifespan (*Figure 5D* inset). By analyzing in detail the kinetics of Ydj1 levels and the appearance of protein aggregates in stochastic simulations (*Figure 5E*) we observed that, due the positive feedback loop inherent to autocatalytic aggregation, once the first protein aggregates appear they rapidly overcome Ydj1 levels by several orders of magnitude, thus making ineffective the relatively small (ca. 50%) increase in Ydj1 levels attained by GAL1p-driven overexpression (*Yahya et al., 2014*).

Finally, we analyzed the interdependencies of Cln3 and Ydj1 in lifespan determination. As shown in *Figure 5F*, overexpression of *CLN3* was able to suppress most of the lifespan reduction of the *ydj1* mutant compared to wild-type cells, these effects being qualitatively similar to those predicted by the integrative model. These data indicate that the molecular deficiencies produced by lack of Ydj1 with regards to lifespan can be greatly corrected by an excess of Cln3, and suggest that this G1 cyclin is a relevant chaperone client involved in cell aging. On the other hand, the effects of *CLN3* overexpression were also clearly attenuated by the *ydj1* deletion, indicating that higher levels of Cln3 require the Ydj1 chaperone to extend lifespan.

## Protein aggregation in young mother cells delays G1 progression and hinders Cln3 function in the nucleus

Given the close temporal relationship observed between the appearance of FFL-GFP aggregates and the final G1 arrest in aging cells, we sought to investigate the effects of protein aggregation on the execution of Start in young cells. Hsp104, Ssa1 and Ydj1 chaperones regulate endogenous prion formation (*Shorter and Lindquist, 2008*). Thus, we fused a synthetic prion-forming domain (PFD) to GFP under the control of a regulatable promoter, and used a non-prion peptide derived from PFD with the same length and amino-acid composition but altered sequence as a control domain (CD) (*Toombs et al., 2012*). While these two peptides displayed similar disorder propensity, they exhibited very distinct prion-like properties (*Figure 6—figure supplement 1A*). PFD, but not CD, produced SDS-resistant high-molecular-weight aggregates as assessed by agarose-gel electrophoresis (*Figure 6—figure supplement 1B*). Moreover, when expressed in young cells, only PFD-mCh formed foci where Ssa1-GFP, and to a much lesser extent Ydj1-GFP, also accumulated (*Figure 6A*). We then analyzed the effects of these synthetic peptides on chaperone mobility by FLIP as above, and found that only PFD expression caused a clear reduction in the mobility of both Ssa1-GFP and Ydj1-GFP (*Figure 6B*), which decreased even further for Ssa1-GFP in cells displaying PFD aggregates (PFD*). These data suggest that PFD expression was able to compromise chaperone availability by sequestering Ssa1 in aggregates with low exchange rates. Next we analyzed the effects of heterologous protein aggregation on the nuclear localization of Cln3, and found that PFD overexpression was sufficient to decrease the nuclear levels of mCtr-Cln3[11A] in a dose-dependent manner (*Figure 6C,D*). Consistent with these results, PFD overexpression increased the average budding size (*Figure 6E*). Sup35 is an endogenous yeast prion that accumulates in the APOD in aging cells (*Saarikangas and Barral, 2015*). Thus, we overexpressed the yeast prion Sup35N domain and observed an increase in the budding volume of cells that showed Sup35N aggregates similar to those with PFD aggregates (*Figure 6E*). In marked contrast, a Sup35N[m3] mutant that does not form aggregates (*Figure 6—figure supplement 1A,B*) did not affect budding volume. More important, overexpression of Cln3 suppressed the increase in budding volume caused by PFD aggregation with

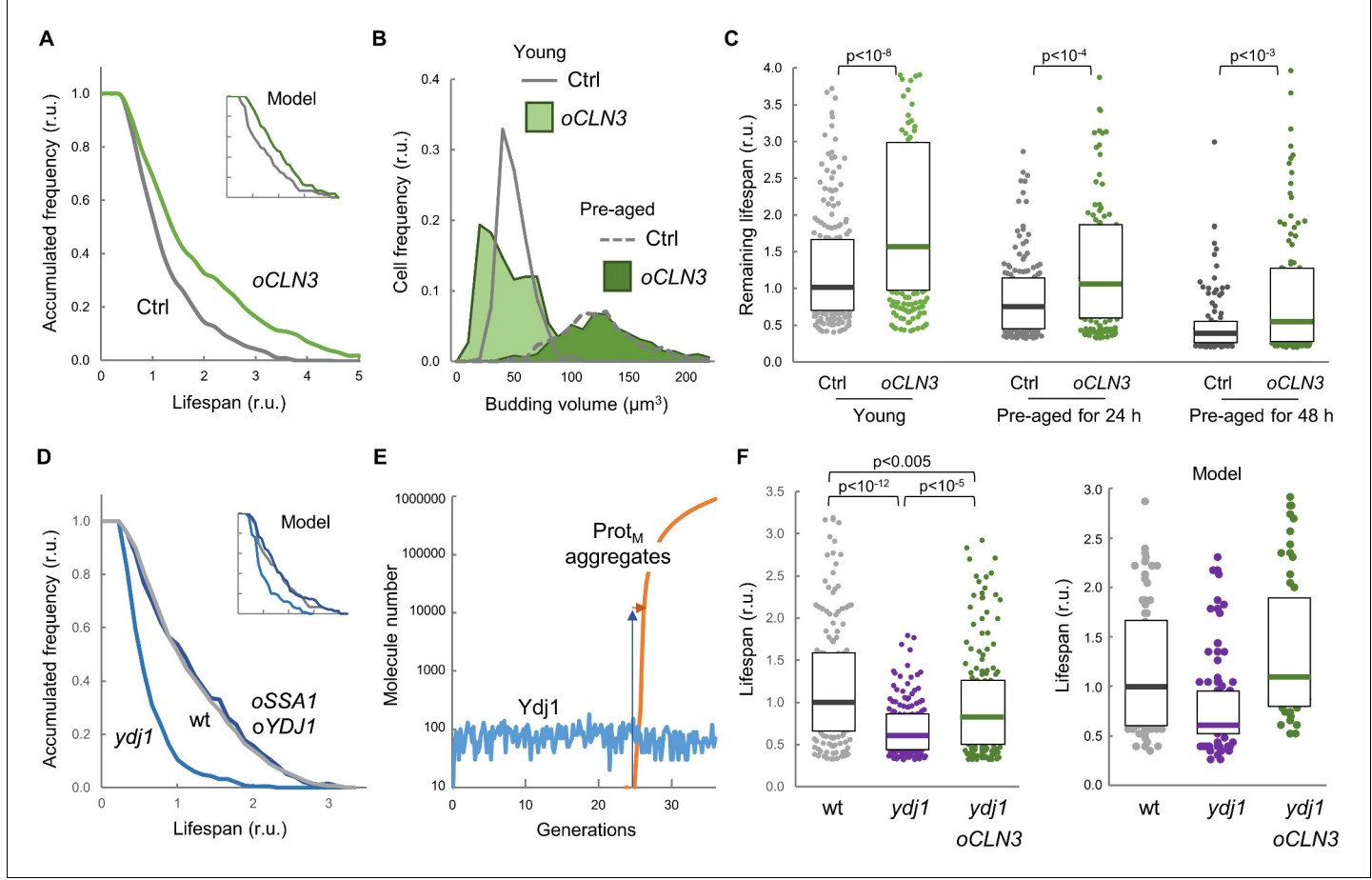

**Figure 5.** Enforced expression of Cln3 increases lifespan in a chaperone-dependent manner. (**A**) Survival curves of control and *CLN3* overexpressing cells (n > 300). Curves predicted by the integrative model in *Figure 4A* are also shown (inset). (**B**) Budding volume distributions (n > 250) of young or old cells pre-aged for 24 hr before induction of *CLN3* expression. (**C**) Lifespan effects of *CLN3* overexpression in young (n = 200) or old cells pre-aged for 24 hr (n = 150) and 48 hr (n = 100). Median ±Q values are also plotted. (**D**) Survival curves of wild-type, Ydj1-deficient and *SSA1 YDJ1* overexpressing cells (n > 300). Curves predicted by the integrative model in *Figure 4A* are also shown (inset). (**E**) Simulation of free Ydj1 and ProtM aggregate numbers during successive replicative cycles in wild-type cells by the integrative model. The results of a representative run in stochastic mode are shown. A 2-order of magnitude increase in Ydj1 levels (blue arrow) would only cause a very small delay (orange arrow) in protein aggregation and, as a consequence, in lifespan. (**F**) Lifespan of wild-type (wt) and Ydj1-deficient cells with empty vector (*ydj1*) or overexpressing *CLN3* (*ydj1 oCLN3*) (n = 150). Median ±Q values are also plotted. The plot on the right shows the predicted lifespan obtained from independent simulations (n = 75) for the corresponding genotypes. Shown p-values were obtained using a Mann-Whitney U test. Results shown in this figure are representative of at least two replicate experiments.

DOI: https://doi.org/10.7554/eLife.48240.019

The following source data and figure supplement are available for figure 5:

**Source data 1.** Enforced expression of Cln3 increases lifespan in a chaperone-dependent manner.
DOI: https://doi.org/10.7554/eLife.48240.021

**Figure supplement 1.** Lifespan analysis by MEP-induced microcolony size.
DOI: https://doi.org/10.7554/eLife.48240.020

no effects on aggregate frequencies (*Figure 6—figure supplement 1C,D*). Finally, since chaperones play important roles in proper coordination of budding size with growth rate (*Ferrezuelo et al., 2012*), we analyzed the possible effects of the accidental presence of PFD aggregates in daughter cells (*Figure 6F*). We observed that the presence of PFD aggregates did not alter ostensibly the average budding size of first-time mother cells, but the dependence on growth rate in G1 was greatly decreased as it had been observed in the *ydj1* mutant (*Ferrezuelo et al., 2012*), further suggesting that PFD aggregation affects Ydj1 availability.

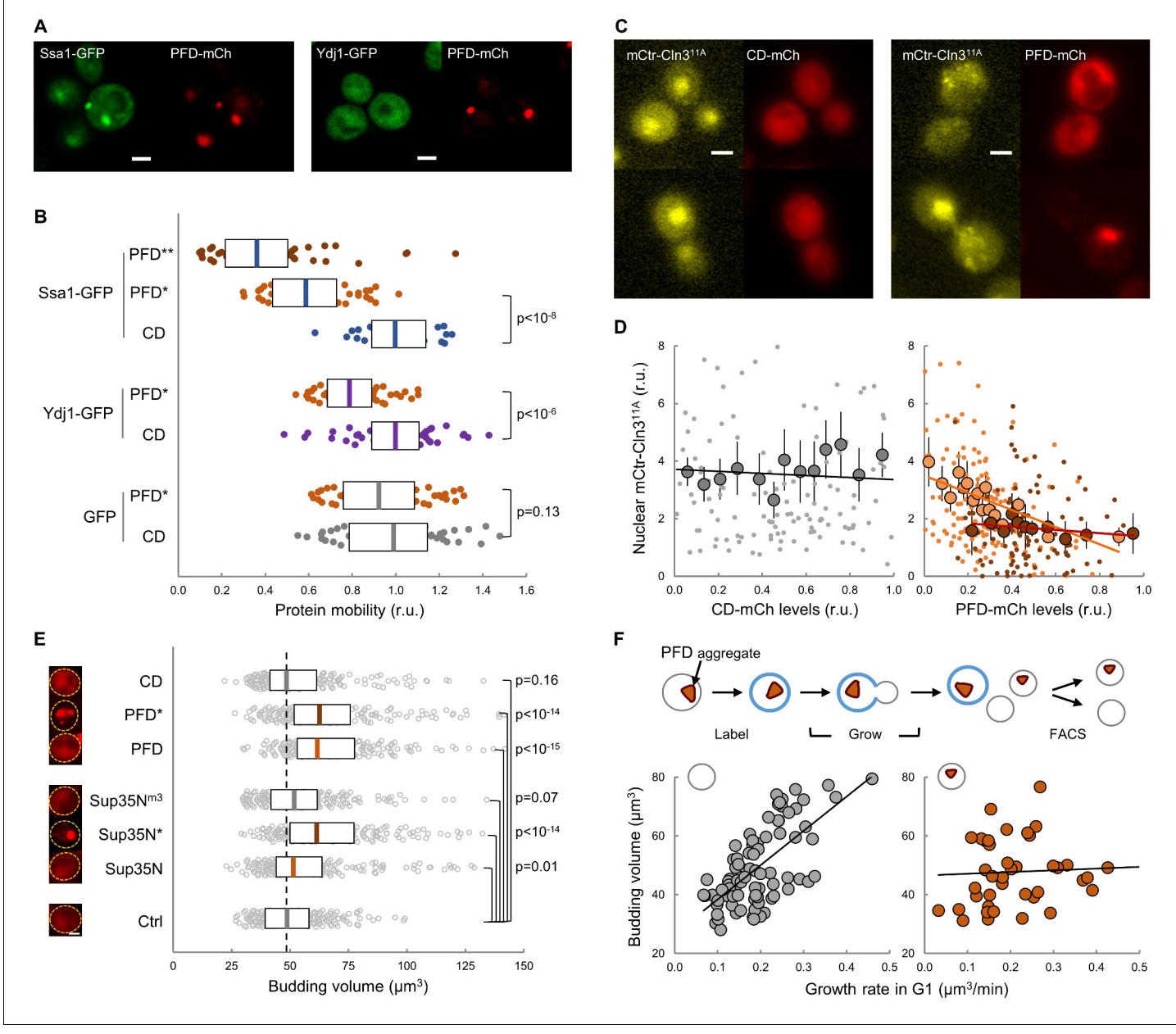

**Figure 6.** Protein aggregation hinders chaperone mobility and nuclear accumulation of Cln3 in young cells. (**A**) Representative young cells expressing the prion-forming domain (PFD)-mCh and either Ssa1-GFP or Ydj1-GFP. (**B**) Mobility of Ssa1-GFP, Ydj1-GFP and GFP in young cells expressing control CD or displaying PFD aggregates (*). Ssa1-GFP mobility was also analyzed within PFD aggregates (**). Median ±Q (n = 50) values are also plotted. (**C**) Representative images of young cells expressing mCtr-Cln3$^{11A}$ and either CD or PFD. (**D**) Nuclear levels of mCtr-Cln3$^{11A}$ in young cells as a function of CD (left) or PFD (right) expression levels. Single-cell (small circles), binned (mean ±CL, n = 10) data and the corresponding linear regression lines are plotted. Cells with PFD aggregates are indicated (red circles). (**E**) Budding volume of young cells expressing the indicated protein domains. Cells with PFD or Sup35N aggregates are indicated (*). Median ±Q (n = 200) values are also plotted. (**F**) Budding volume of newborn daughter cells in the absence (left, n = 82) or presence (right, n = 42) of PFD aggregates after FACS selection as a function of growth rate in G1. Shown p-values were obtained using a Mann-Whitney U test. Bar = 2 μm. Results shown in this figure are representative of at least two replicate experiments.
DOI: https://doi.org/10.7554/eLife.48240.022

The following source data and figure supplement are available for figure 6:

**Source data 1.** Protein aggregation hinders chaperone mobility and nuclear accumulation of Cln3 in young cells.
DOI: https://doi.org/10.7554/eLife.48240.024

**Figure supplement 1.** Enforced protein aggregation and *CLN3* overexpression effects in young cells.
DOI: https://doi.org/10.7554/eLife.48240.023

## Lifespan shortening by protein aggregation is suppressed by overexpression of chaperones or Cln3

To confirm the notion that proteotoxic aggregates limit replicative lifespan we expressed the above-mentioned synthetic peptides in wild-type cells in the CLiC microfluidics chamber. Notably, PFD caused a dramatic decrease in lifespan, which was accentuated even more in mother cells showing PFD aggregates (*Figure 7A*). The frequency of cells in G1 at death also increased about 4-fold relative to young mother cells (*Figure 7B*), and there was a strong correlation between PFD concentration and the occurrence of death in the following 180 min (*Figure 7C*). By contrast, CD levels did not correlate at all with the timing of cell death (*Figure 7D*). Moreover, as previously observed with FFL-GFP in aging cells, PFD aggregation increased Hsp104 levels in young cells (*Figure 7—figure supplement 1A*). However, different from aging cells, PFD and Hsp104 foci colocalized in young cells. Finally, budding size in PFD-expressing mother cells was larger in successive divisions compared to CD-expressing cells (*Figure 7—figure supplement 1B*), indicating longer delays in G1.

Our data point to the notion that premature protein aggregation shortens replicative lifespan by compromising chaperone availability which, in turn, would hinder nuclear accumulation of cyclin Cln3 and progressively delay Start, leading the cell to an irreversible G1 arrest and death. To test this possibility further, we decided to analyze the effects of enforced chaperone or Cln3 expression in the lifespan of PFD-expressing cells. Notably, as predicted by our model (*Figure 7E*), we found that overexpression of *SSA1* and *YDJ1* from the dual *GAL1-10* promoter partially suppressed the lifespan reduction caused by PFD (*Figure 7F*), the lifespan being even closer to control CD-expressing cells when the copy number of seven chaperone genes (*SSA1*, *YDJ1*, *HSP82*, *CDC37*, *CDC48*, *UFD1*, *NPL4*) that cooperate in ER-release and proper Cdk-cyclin complex formation was duplicated (*Figure 7G*). Finally, also as predicted by the model (*Figure 7E*), the lifespan was totally comparable to control cells when PFD-expressing cells were subject to *CLN3* overexpression (*Figure 7H*). These results give additional support to the notion that protein aggregation in young cells leads to a premature G1 arrest by specifically inhibiting chaperone- and G1 cyclin-dependent execution of Start.

## Discussion

It is generally accepted that aging cells undergo many different deleterious processes that somehow restrain proliferation and ultimately lead to cell death. However, their specific relevance and cause-effect relationships are just starting to emerge. Here we show that most yeast cells arrest in G1 before death and display low nuclear levels of cyclin Cln3, a key activator of Start that is particularly sensitive to chaperone status (*Moreno et al., 2019*; *Parisi et al., 2018*; *Vergés et al., 2007*). By using several independent approaches, we show that chaperone availability is seriously compromised in aged cells, and we find that blockade of cell-cycle entry finely correlates with the appearance of visible aggregates of a chaperone client reporter. A mathematical model integrating the role of chaperones in proteostasis and cyclin Cln3 activation is able to recapitulate our observations in aging cells. Notably, overexpression of Cln3 increases lifespan in a chaperone-dependent manner. As also predicted by the model, overexpression of aggregation-prone proteins in young cells decreases chaperone availability and restrains nuclear accumulation of Cln3 and, hence, cell-cycle entry. Finally, lifespan shortening by enforced protein aggregation can be suppressed by increased expression of specific chaperones or cyclin Cln3. Overall, these data establish a molecular mechanism linking loss of protein homeostasis to proliferation arrest in aged yeast cells.

Our data agree with the recent observation that expression of G1/S genes is greatly compromised in aged cells (*Neurohr et al., 2018*), concurrently with an increase in the nuclear levels of Whi5. Since Whi5 is phosphorylated and exported to the cytoplasm by G1 Cdk-cyclin complexes (*de Bruin et al., 2004*; *Costanzo et al., 2004*), our observation that nuclear accumulation of mCtr-Cln3[11A] is hindered in aging cells would explain, at least in part, the increase in nuclear Whi5 and the deficiencies in the activation of the G1/S regulon. Cells lacking Cln3 display a dramatic delay in G1, but do not arrest at Start unless *CLN1* and *CLN2* are disrupted (*Richardson et al., 1989*). Thus, the observed final G1 arrest should also involve other mechanisms restraining Cln1/2 levels or activity. It has been recently proposed (*Neurohr et al., 2018*) that the accumulation of ERCs in aged cells (*Shcheprova et al., 2008*; *Sinclair and Guarente, 1997*) could have a direct inhibitory role on the *CLN2* promoter at the nuclear pore (*Kumar et al., 2018*). Interestingly, ERCs increase their levels around the time when G1 progression displays a clear delay (*Morlot et al., 2019*). As an alternative

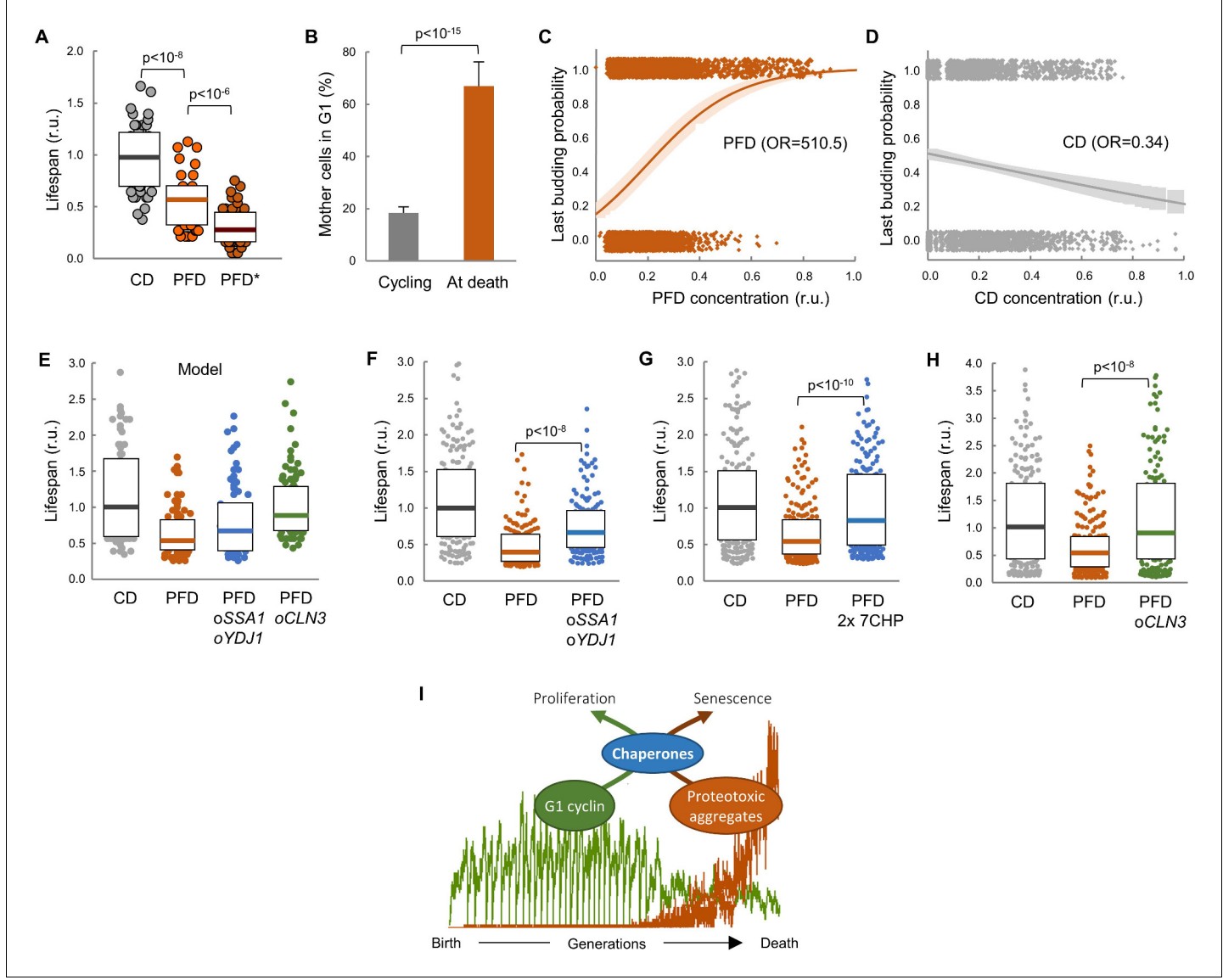

**Figure 7.** Lifespan shortening by protein aggregation can be overcome by enforced expression of chaperones or Cln3. (A) Lifespan effects of CD or PFD expression in young cells in the CLiC microfluidics chamber (n = 50). Cells with PFD aggregates at the initial time point are indicated (*). Median ±Q values are also plotted. (B) Percentage (±CL, n = 57) of PFD expressing cells in G1 in young cycling cultures or at death. (C–D) Last budding probability within the following 180 min (twice the generation time of young mother cells) after reading PFD (C) or CD (D) cell concentration. Sampled single-cell data (closed circles) and the logistic regression lines (mean ±CL) are plotted. Odds ratios are also indicated. (E) Predicted lifespan effects of *SSA1 YDJ1* overexpression and *CLN3* overexpression in PFD expressing cells (n = 75). Predicted values for control CD-expressing cells are shown as reference. Median ±Q values are also plotted. (F–H) Lifespan effects of concerted *SSA1 YDJ1* overexpression (F), duplication of seven chaperone genes (2 × 7CHP: *SSA1*, *YDJ1*, *HSP82*, *CDC37*, *CDC48*, *UFD1* and *NPL4*) (G) or *CLN3* overexpression (H) in PFD expressing cells (n > 150). Values of control CD-expressing cells are shown as reference. Median ±Q values are also plotted. (I) By compromising chaperone availability, proteostasis deterioration would exclude cyclin Cln3 from the nucleus and, as a direct consequence, drive the cell into senescence. Shown p-values were obtained using a Mann-Whitney U test. Results shown in this figure are representative of at least two replicate experiments.

DOI: https://doi.org/10.7554/eLife.48240.025

The following source data and figure supplement are available for figure 7:

**Source data 1.** Lifespan shortening by protein aggregation can be overcome by enforced expression of chaperones or Cln3.
DOI: https://doi.org/10.7554/eLife.48240.027

**Figure supplement 1.** Enforced protein aggregation increases levels of Hsp104 and budding size during aging.
DOI: https://doi.org/10.7554/eLife.48240.026

view, since Cln2 interacts with Ssa1,2 chaperones (*Gong et al., 2009*) and likely requires chaperoning activities similar to Cln3 (*Ferrezuelo et al., 2012*; *Moreno et al., 2019*), our observations on Cln3 could also apply to basal levels of Cln1 and Cln2 and, hence, contribute to explaining the final G1 arrest.

Mitochondrial membrane potential has been shown to play a key role in dissolution of protein aggregates (*Ruan et al., 2017*; *Zhou et al., 2014*) and loss of membrane potential correlates with the SEP in a fraction of aging cells (*Fehrmann et al., 2013*). On the other hand, vacuolar acidity declines early during aging, and conditions that prevent this decline ameliorate mitochondrial function and extend lifespan (*Hughes and Gottschling, 2012*). Related to this, the vacuolar protein Vac17 has been shown to be involved in asymmetric segregation of protein aggregates (*Hill et al., 2016*). Interestingly, we have observed that enforced protein aggregation in young cells increased vacuolar pH (our unpublished observations). These findings support the notion that mitochondrial defects, vacuolar dysfunction and accumulation of protein aggregates during aging would exhibit multiple functional interactions (*Hill et al., 2017*), cell death being the result of many intertwined defects in key cellular processes. In addition, cells that keep growing while cell-cycle arrested undergo cytoplasmic dilution (*Neurohr et al., 2019*), eventually contributing to a general breakdown of cellular homeostasis.

In our experiments with the CLiC microfluidics chamber, a fraction of cells (ca. 30%) were not arrested in G1 at death and showed a slightly shorter lifespan, which agrees with published findings (*Delaney et al., 2013*). PFD-overexpressing cells also displayed a similar percentage of death outside G1. Interestingly, enforced aggregation of the Rnq1 yeast prion causes a G2/M arrest with monopolar spindles (*Treusch and Lindquist, 2012*). In all, these observations suggest that proteostasis defects would also hinder cell-cycle progression and limit replicative lifespan after bud emergence.

Ydj1 cooperates with Hsp104 and Hsp70 chaperones to recycle misfolded proteins (*Glover and Lindquist, 1998*), and improper recruitment of chaperones to misfolded proteins has very negative effects in lifespan (*Hanzén et al., 2016*). In addition, Ydj1 associates with the APOD playing a key role in its asymmetric segregation to the mother compartment during cell division (*Saarikangas and Barral, 2015*; *Saarikangas et al., 2017*). Ydj1-deficient cells are particularly short-lived (*Hill et al., 2014*), supporting the relevance of proteostasis mechanisms in lifespan determination. In agreement with this notion, overexpression of Hsp104 restores proteasome activity in aging cells (*Andersson et al., 2013*) and suppresses lifespan defects of *sir2* mutants (*Erjavec et al., 2007*). However, enforced expression of Hsp104 did not increase lifespan significantly in wild-type cells (*Andersson et al., 2013*). We obtained similar results by concurrent overexpression of Ssa1 and Ydj1. Albeit surprisingly, our model predicted that chaperone overexpression would have a very limited effect on lifespan. This output of the model would be explained by (1) the positive feedback loop whereby proteostasis decline is accelerated by the autocatalytic accumulation of aggregated proteins (*Andersson et al., 2013*), and (2) the fact that client/chaperone ratios in molecule numbers per cell are intrinsically overwhelming. Nonetheless, the inability of chaperone overexpression to extend lifespan would also underscore the existence of chaperone-independent mechanisms affecting execution of Start in aged cells as above mentioned, such as ERC-mediated downregulation of G1/S genes (*Neurohr et al., 2018*). On the other hand, *CLN3* overexpression suppresses the G1 progression delay of Ydj1-deficient cells with no effects on the expression of related chaperones (*Vergés et al., 2007*), and we show here that high levels of Cln3 rescue the short lifespan of Ydj1-deficient cells. These data give support to the downstream role of Cln3 with respect to chaperones in the proteotoxic aggregation pathway. This interplay between Cln3 and protein aggregates through chaperone availability is simulated in *Figure 7I*, where free Cln3 levels decay in aged cells as protein aggregates increase in an autocatalytic loop.

Aged cells display increased levels of expression of many genes of the environmental stress response (ESR) (*Janssens et al., 2015*; *Yiu et al., 2008*), which suggests that cells would sense proteostasis defects during aging similarly to other stress instances in which chaperone availability is assumed to be temporarily compromised (*de Nadal et al., 2011*). Since many ESR genes are also upregulated in G0 cells obtained by nutrient starvation (*Gasch et al., 2000*), expression similarities between G0 and aged cells would be due to the ESR signature. However, *RIM15*, which is strongly upregulated by nutrient starvation but not by heat stress, is also upregulated in aged cells, indicating that G0 and aged cells might activate common transcriptional programs other than the ESR.

Proteostasis defects have been associated with cell aging in many different model organisms (*Klaips et al., 2018*), and the key factors that maintain the proteome in a conformationally-active state are exquisitely conserved. On the other hand, as in yeast, human G1 Cdk-cyclin complexes require the participation of chaperones also involved in general proteostasis (*Diehl et al., 2003*; *Hallett et al., 2017*). Thus, we foresee that mechanisms similar to those shown here for yeast cells could also play a prominent role in restraining proliferation in aging human cells.

# Materials and methods

## Key resources table

| Reagent type or resource | Designation | Source or reference | Identifiers | Additional information |
|---|---|---|---|---|
| Strain, strain background (*Saccharomyces cerevisiae*) | BY4741 | Lab stock | | *MATa his3-Δ1 leu2Δ0 met 15Δ0 ura3Δ0, from S288C* |
| Strain, strain background (*S. cerevisiae*) | MAG248 | This work | | *MATa his3-Δ1 leu2Δ0 met15Δ0 ura3Δ0 NAT::TEFp-GFP, from S288C* |
| Strain, strain background (*S. cerevisiae*) | MAG261 | (*Moreno et al., 2019*) | | *MATa his3-Δ1 leu2Δ0 met15Δ0 ura3Δ0 YDJ1-GFP-FS::HIS3, from S288C* |
| Strain, strain background (*S. cerevisiae*) | MAG1078 | This work | | *MATa his3-Δ1 leu2Δ0 met15Δ0 ura3Δ0 YDJ1-GFP-FS::HIS3 SSA1-mCherry::HYG, from S288C* |
| Strain, strain background (*S. cerevisiae*) | MAG1689 | This work | | *MATa his3-Δ1 leu2Δ0 met15Δ0 ura3Δ0 SSA1-GFP::HIS3 OLE1-mCherry::GEN, from S288C* |
| Strain, strain background (*S. cerevisiae*) | YOR083W-GFP | Lab stock | | *MATa his3-Δ1 leu2Δ0 met15Δ0 ura3Δ0 WHI5-yGFP::HIS3, from S288C* |
| Strain, strain background (*S. cerevisiae*) | CML128 | (*Gallego et al., 1997*) | | *MATa leu2-3,112 ura3-52 trp1-1 his4-1 can<sup>r</sup>, from 1788* |
| Strain, strain background (*S. cerevisiae*) | MAG1077 | This work | | *MATa leu2-3,112 ura3-52 trp1-1 his4-1 can<sup>r</sup> WHI5-sGFP::GEN HSP104-mCherry::HYG, from 1788* |
| Strain, strain background (*S. cerevisiae*) | MAG1512 | (*Moreno et al., 2019*) | | *MATa leu2-3,112 ura3-52 trp1-1 his4-1 can<sup>r</sup> NAT::TEF1p-mCherry, from 1788* |
| Strain, strain background (*S. cerevisiae*) | MAG1767 | This work | | *MATa leu2-3,112 ura3-52 trp1-1 his4-1 can<sup>r</sup> mCitrine-CLN3(11A)::NAT, from 1788* |
| Strain, strain background (*S. cerevisiae*) | MAG1767 | This work | | *MATa leu2-3,112 ura3-52 trp1-1 his4-1 can<sup>r</sup> HSP104-mCherry::HYG, from 1788* |
| Strain, strain background (*S. cerevisiae*) | UCC5179 | (*Lindstrom and Gottschling, 2009*) | | *MATa ade2::hisG his3 leu2 lys2 ura3Δ0 trp1Δ63 hoΔ::SCW11pr-Cre-EBD78-NatMX loxP-UBC9-loxP-LEU2 loxP-CDC20-intron-loxP-HPHMX, from S288C* |
| Strain, strain background (*S. cerevisiae*) | MAG1013 | This work | | *MATa ade2::hisG his3 leu2 lys2 ura3Δ0 trp1Δ63 hoΔ::SCW11pr-Cre-EBD78-NatMX loxP-UBC9-loxP-LEU2 loxP-CDC20-intron-loxP-HPHMX (ARS-CEN URA3 HSP104 SSA1 YDJ1 HSC82 CDC37 CDC48 UFD1 NPL1), from S288C* |

*Continued on next page*

*Continued*

| Reagent type or resource | Designation | Source or reference | Identifiers | Additional information |
|---|---|---|---|---|
| Strain, strain background (*S. cerevisiae*) | MAG1095 | This work | | *MATa ade2::hisG his3 leu2 lys2 ura3Δ0 trp1Δ63 hoΔ::SCW11 pr-Cre-EBD78-NatMX loxP-UBC9-loxP-LEU2 loxP-CDC20-intron-loxP-HPHMX YDJ1-GFP-FS::HIS3, from S288C* |
| Strain, strain background (*S. cerevisiae*) | MAG1096 | This work | | *MATa ade2::hisG his3 leu2 lys2 ura3Δ0 trp1Δ63 hoΔ::SCW11pr-Cre-EBD78-NatMX loxP-UBC9-loxP-LEU2 loxP-CDC20-intron-loxP-HPHMX SSA1-GFP::HIS3, from S288C* |
| Strain, strain background (*S. cerevisiae*) | MAG1578 | This work | | *MATa ade2::hisG his3 leu2 lys2 ura3Δ0 trp1Δ63 hoΔ::SCW11 pr-Cre-EBD78-NatMX loxP-UBC9 -loxP-LEU2 loxP-CDC20-intron -loxP-HPHMX ydj1Δ::GEN, from S288C* |
| Strain, strain background (*S. cerevisiae*) | MAG1745 | This work | | *MATa ade2::hisG his3 leu2 lys2 ura3Δ0 trp1Δ63 hoΔ:: SCW11pr-Cre-EBD78-NatMX loxP-UBC9-loxP-LEU2 loxP-CDC20-intron-loxP- HPHMX YDJ1-GFP-FS::HIS3 SSA1-mCherry::KAN, from S288C* |
| Strain, strain background (*S. cerevisiae*) | MAG1952 | This work | | *MATa ade2::hisG his3 leu2 lys2 ura3Δ0 trp1Δ63 hoΔ:: SCW11pr-Cre-EBD78-NatMX loxP-UBC9-loxP-LEU2 loxP-CDC20-intron-loxP-HPHMX Hsp104-mCherry::GEN, from S288C* |
| Strain, strain background (*S. cerevisiae*) | MAG2060 | This work | | *MATa ade2::hisG his3 leu2 lys2 ura3Δ0 trp1Δ63 hoΔ:: SCW11pr-Cre-EBD78-NatMX loxP-UBC9-loxP-LEU2 loxP-CDC20-intron-loxP-HPHMX GAL1p-CLN3 URA3::TRP1, from S288C* |
| Strain, strain background (*S. cerevisiae*) | MAG1253 | This work | | *MATa ade2::hisG his3 leu2 lys2 ura3Δ0 trp1Δ63 hoΔ:: SCW11pr-Cre-EBD78-NatMX loxP-UBC9-loxP-LEU2 loxP-CDC20-intron-loxP-HPHMX trp1Δ63::SCW11pr-Cre-EBD78 -KanMX4, from S288C* |
| Strain, strain background (*S. cerevisiae*) | MAG1569 | This work | | *MATa ade2::hisG his3 leu2 lys2 ura3Δ0 trp1Δ63 hoΔ:: SCW11pr-Cre-EBD78-NatMX loxP-UBC9-loxP-LEU2 loxP-CDC20-intron-loxP-HPHMX trp1Δ63::SCW11pr-Cre-EBD78- KanMX4 CLB2-GFP::HIS3MX, from S288C* |
| Strain, strain background (*S. cerevisiae*) | MAG1576 | This work | | *MATa ade2::hisG his3 leu2 lys2 ura3Δ0 trp1Δ63 hoΔ:: SCW11pr-Cre-EBD78-NatMX loxP-UBC9-loxP-LEU2 loxP-CDC20-intron-loxP-HPHMX trp1Δ63::SCW11pr-Cre- EBD78-KanMX4 CLB2-GFP::HIS3 GALp-CLN3 -URA3::TRP1, from S288C* |

*Continued on next page*

*Continued*

| Reagent type or resource | Designation | Source or reference | Identifiers | Additional information |
|---|---|---|---|---|
| Strain, strain background (*S. cerevisiae*) | MAG1795 | This work | | *MATa ade2::hisG his3 leu2 lys2 ura3Δ0 trp1Δ63 hoΔ:: SCW11pr-Cre-EBD78-NatMX loxP-UBC9-loxP-LEU2 loxP-CDC20-intron-loxP-HPHMX trp1Δ63::SCW11pr-Cre-EBD78 -KanMX4 CLB2-GFP::HIS3MX Δcln3::URA3MX, from S288C* |
| Strain, strain background (*S. cerevisiae*) | W303-1A | Lab stock | | *MATa ade2-1 trp1-1 leu2-3,111 his3-11,75 ura3 can1-100, from W303* |
| Strain, strain background (*S. cerevisiae*) | KSY083-5 | (*Schmoller et al., 2015*) | | *MATa ADE2 trp1-1 leu2-3,111 his3-11,75 ura3 can1-100 mCitrine-CLN3-11A::NAT* |
| Strain, strain background (*S. cerevisiae*) | MAG876 | This work | | *MATa ade2-1 trp1-1 leu2-3, 111 his3-11,75 ura3 can1- 100 SSA1-GFP::HIS3, from W303* |
| Recombinant DNA reagent | YCplac22 | Lab stock | | Centromeric *TRP1* vector |
| Recombinant DNA reagent | YCplac33 | Lab stock | | Centromeric *URA3* vector |
| Recombinant DNA reagent | YCpGAL | Lab stock | | *GAL1/10* p in YCplac22 |
| Recombinant DNA reagent | p425MET25-FFL-GFP | (*Abrams and Morano, 2013*) | | *MET25p-FFL-GFP* in pRS425 |
| Recombinant DNA reagent | pCYC87 | Lab stock | | *GAL1/10p-CLN3-3HA* in YCplac33 |
| Recombinant DNA reagent | pMAG438 | (*Moreno et al., 2019*) | | *SSA1 YDJ1 HSC82 CDC37 CDC48 UFD1 NPL4* in YAC *URA3* |
| Recombinant DNA reagent | pMAG600 | This work | | *GAL1/10p-Sup35Nm3- GFP* in YCplac22 |
| Recombinant DNA reagent | pMAG602 | This work | | *GAL1/10p-PFD-GFP* in YCplac22 |
| Recombinant DNA reagent | pMAG604 | This work | | *GAL1/10p-CD-GFP* in YCplac22 |
| Recombinant DNA reagent | pMAG605 | This work | | *GAL1/10p-Sup35N- GFP* in YCplac22 |
| Recombinant DNA reagent | pMAG610 | This work | | *GAL1/10p-GFP* in YCplac22 |
| Recombinant DNA reagent | pMAG633 | This work | | *GAL1/10p-PFD-mCh* in YCplac33 |
| Recombinant DNA reagent | pMAG634 | This work | | *GAL1/10p-CD-mCh* in YCplac33 |
| Recombinant DNA reagent | pMAG1182 | This work | | *GAL1p-SSA1 GAL10p- YDJ1* in YCplac33 |
| Recombinant DNA reagent | pMAG1228 | Lab stock | | *TE1Fp-GFP* in YCplac33 |
| Antibody | αGFP (Mouse monoclonal) | Merck | G1546 | |
| Chemical compound, drug | Azetidine-2-carboxilic acid | Sigma-Aldrich | A0760 | |
| Chemical compound, drug | β-estradiol | Sigma-Aldrich | E2758 | |

*Continued on next page*

*Continued*

| Reagent type or resource | Designation | Source or reference | Identifiers | Additional information |
|---|---|---|---|---|
| Software, algorithm | MODEL1901210001 | This work | | BioModels database |
| Software, algorithm | ImageJ | Wayne Rasband, NIH | | imagej.nih.gov/ij/download.html |
| Software, algorithm | BudJ | (*Ferrezuelo et al., 2012*) | | ibmb.csic.es/groups/spatial-control-of-cell-cycle-entry |
| Software, algorithm | CoinRICSJ | (*Moreno and Aldea, 2019*) | | ibmb.csic.es/groups/spatial-control-of-cell-cycle-entry |
| Software, algorithm | RICS analysis plugins | Jay Unruh, Stowers Institute | | research.stowers.org/imagejplugins |
| Software, algorithm | Microcolony_size.ijm | This work | | ibmb.csic.es/groups/spatial-control-of-cell-cycle-entry |
| Software, algorithm | PAPA | (*Toombs et al., 2012*) | | combi.cs.colostate.edu/supplements/papa |

## Strain constructions and growth conditions

Parental strains and methods used for chromosomal gene transplacement and PCR-based directed mutagenesis have been described (*Ferrezuelo et al., 2012*). Unless stated otherwise, all gene fusions in this study were expressed at endogenous levels at their respective loci. As C-terminal fusion of GFP or other tags has strong deleterious effects on Ydj1 function, we inserted GFP at amino acid 387, between the dimerization domain and the C-terminal farnesylation sequence of Ydj1. This construct had no detectable effects on growth rate or cell volume when expressed at endogenous levels (*Saarikangas et al., 2017*). The mCitrine-Cln3$^{11A}$ fusion protein contains a hypo-active and hyperstable cyclin with 11 amino acid substitutions (R108A, T420A, S449A, T455A, S462A, S464A, S468A, T478A, S514A, T517A, T520A) that allows its detection by fluorescence microscopy with no gross effects on cell cycle progression (*Schmoller et al., 2015*). Centromeric plasmids and yeast artificial chromosomes containing chaperone genes were obtained by multiple-fragment recombination (*Moreno et al., 2019*). Cells were grown for 7–8 generations in SC medium with 2% glucose at 30°C unless stated otherwise. GAL1p-driven gene expression was induced by addition of 2% galactose to cultures grown in 2% raffinose at OD600 = 0.5. 1 μM β-estradiol was used to activate the Mother Enrichment Program (MEP) as described (*Lindstrom and Gottschling, 2009*). Azetidine 2-carboxylic acid (AZC) was used at 10 mM.

## Time-lapse microscopy

Cells were analyzed by time-lapse microscopy within the CLiC microfluidic device as described (*Fehrmann et al., 2013*) in SC-based media at 30°C essentially as described (*Ferrezuelo et al., 2012*) using a fully-motorized Leica AF7000 microscope with a 63X/1.3NA oil-immersion objective. The media was pumped into the microfluidic device at a rate of 20 μL/min. Time-lapse images were taken every 10 min. Time-lapse images were analyzed with the aid of BudJ (*Ferrezuelo et al., 2012*), an ImageJ (Wayne Rasband, NIH) plugin that can be obtained from ibmb.csic.es/groups/spatial-control-of-cell-cycle-entry to obtain cell dimensions and fluorescence data as described (*Ferrezuelo et al., 2012*); budding events were identified visually. Wide-field microscopy is able to collect the total fluorescence emitted by yeast cells and, consequently, cellular concentration of fluorescent fusion proteins was obtained by dividing the integrated fluorescence signal within the projected area of the cell by its volume. The nuclear compartment was delimited as described (*Ferrezuelo et al., 2012*). Briefly, the gravity center from brightest pixels in the cell was used as center of a projected circle with area equal to that expected for the nucleus (17% of the cell projected area). Since the signal in the nuclear projected area is influenced by both nuclear and cytoplasmic fluorescence, determination of the nuclear concentration required specific calculations as described (*Moreno et al., 2019*). Intracellular foci were detected with BudJ as pixels with a fluorescence value above a certain threshold relative to the median cell fluorescence that produced a contiguous area with a minimum size (both set by the user). In a typical set up, pixels were selected if at least 30%

brighter than the cell median, with a minimal size of 0.4 µm. Photobleaching during acquisition was negligible (less than 0.1% per time point) and autofluorescence was always subtracted.

## Biotin labeling of the cell wall for aged cells detection

MEP-derived cells were labeled with Sulfa-NHS-LC-Biotin (Pierce) as described (*Lindstrom and Gottschling, 2009*), and seeded in SC medium with 1 µM β-estradiol. After ageing for 1 or 2 days, cells were collected using a 0.2 µm pore centrifuge filter and a soft spin. Cells were washed twice with PBS in the column and stained with a Streptavidin-APC conjugate solution (2 µg/ml in PBS) for 30 min at 4°; simultaneously, bud scars were labeled with a compatible WGA-conjugated fluorochrome at 20 µg/ml. Afterwards cells were washed twice with media and transferred to 35 mm glass-bottom culture dishes (GWST-3522, WillCo) before microscopy.

## Chaperone mobility analysis by FLIP and RICS

We used fluorescence loss in photobleaching (FLIP) to analyze chaperone mobility in a Zeiss LSM780 confocal microscope with a 40X/1.2NA water-immersion objective at room temperature. FLIP was used as a qualitative assay to determine Ssa1-GFP and Ydj1-GFP mobility in the whole cell. A small circular region of the cytoplasm (3.6 µm$^2$) was repetitively photobleached at full laser power while the cell was imaged at low intensity every 0.5 s to record fluorescence loss. After background subtraction, fluorescence data from an unbleached cell region were made relative to the initial time point, and a mobility index was calculated as the inverse of the fluorescence half-life obtained by fitting an exponential function. We noticed a clear dependency of this mobility index on cell size. Using a dataset of cells with very wide size range and expressing free GFP, we obtained an expression to correct the mobility index for cell size with $MI_c = MI \cdot r^{1.49}$ where $MI$ is the raw mobility index, $r$ is the cell radius and $MI_c$ is the corrected mobility index (*Figure 2—figure supplement 2*).

Raster Image Correlation Spectroscopy (RICS) analysis was performed in a Zeiss LSM780 confocal microscope with a 63X/1.3NA water-immersion objective; specifically, we used a 35 nm pixel size and 12.6µs dwell time, tacking a stack of 100 frames (2 s/frame) in photon-counting detection mode, at room temperature. To obtain the coefficient of diffusion, 64 × 64 pixel stacks were used to remove the immobile fraction with a five frame moving average and analyzed using a set of plugins written by Jay Unruh (Stowers Institute) for ImageJ. The resulting autocorrelation function (ACF) in the scanning direction, that is $G_{RICS}(\xi,0)$, for each cell was obtained and averaged for each group of cells to determine the diffusion coefficient (D) and the number of molecules in the focus (N) by fitting the data to a simple diffusion model. In order to assess the uncertainty of the predicted D values from pooled RICS data by non-linear regression methods we used a Monte Carlo approximation.

Coincidence analysis with RICS data was carried out with CoinRICSJ (*Moreno and Aldea, 2019*). Briefly, after removal of the immobile fraction as described above, the ACF of each pixel was obtained using a 16 pixel range only in the raster direction. The intercept obtained by linear regression of the ACF (no specific model of diffusion assumed) was used as an approximation of the inverse of the number of moving particles (N). Then, the fluorescence intensity (I) at each pixel was used calculate the brightness (B) parameter as B = I/N, which were assembled into B maps covering the whole image being analyzed. Finally, correlation between Bmaps was analyzed using the Pearson's correlation coefficient, setting the threshold as the mean value in the B map. These correlation coefficients assess the degree of spatiotemporal coincidence of moving particles of the two proteins analyzed as a function of the number of fluorescent molecules per particle (*Moreno and Aldea, 2019*).

## Lifespan analysis by MEP-induced microcolony size

MEP strains were grown as above, diluted to OD600 = 0.01, and plated in 500 µL at ~3·10$^4$ cfu/cm$^2$ onto 35 mm 2% agar plates containing SC medium with 2% glucose or 2% galactose and 1 µM β-estradiol. Once the plates were dry, they were incubated at 30°C for 4 days. Finally, the microcolonies were imaged using a Leica AF7000 microscope with a 20X/0.5NA dry objective. As a proof of concept for the method, we measured the microcolony size produced by cells pre-aged in liquid media with 1 µM β-estradiol for increasing amounts of time, and we observed a progressive decrease in microcolony size as a function of pre-aging time in liquid media before plating (*Figure 5—figure supplement 1C*). The microcolony area was determined semiautomatically using an

ImageJ macro (microcolony_size.ijm). Briefly, after thresholding and binarization, segmentation of adjacent microcolonies with the watershed function, and exclusion of the objects at the edge of the image, the area of particles (holes included) was measured. Microcolonies that were too small (with less than 4–5 cell bodies) or too big (microcolonies where cells had likely escaped from the MEP) were filtered out.

## Integrative mathematical model

The wiring diagram used to describe the interaction between Start and the protein folding/aggregation pathway is described in *Figure 4—figure supplement 1*. We chose to focus only upon execution of Start because experimentally we found that over 75% of aging cells arrest in G1 before death. To simulate the rest of the cell cycle we run a fixed timer. The Start network was also simplified to a constantly diluting (*Schmoller et al., 2015*) Whi5 molecule that is phosphorylated and inactivated by fluctuating Cln3 (*Liu et al., 2015*), which requires the concerted action of Ssa1 and Ydj1 chaperones for full activation (*Vergés et al., 2007*). Thus, execution of Start was modeled to take place when a minimal Whi5 threshold was reached. *Figure 4—figure supplement 1* also details the wiring diagram used to describe the protein aggregation pathway. This again is a simplified approximation and includes dimers (which are assumed to represent a pool of all non-nucleated oligomers) and hexamers (which are assumed to represent a pool of all nucleated oligomers). This drastically decreases the number of species in the model and the complexity of the system, and we specifically chose hexamers to represent the nucleated form as they appear to have a critical size for stabilization of the oligomer in aggregating proteins (*Breydo and Uversky, 2015*; *Xue et al., 2008*). The Hsp104 disaggregase is required for disassembling large aggregates and works in conjunction with Ssa1 and Ydj1 chaperones (*Okuda et al., 2015*). It is therefore included in the dissociation of nucleated aggregates (hexamers), but not in monomer or dimer refolding. Ssa1 and Ydj1 are also able to suppress aggregation, presumably by refolding monomers and other small oligomers that are not nucleated, so we allow the Ssa1/Ydj1 chaperones to bind and refold these states. It is assumed that refolding always adds to the folded protein pool and that chaperones are not released until the misfolded proteins are either degraded or obtain their correct conformation.

The wiring diagram was converted into a model using COPASI (*Hoops et al., 2006*) with chemical reactions shown in *Supplementary file 1*. All chemical reactions are assumed to follow mass-action kinetics and, hence, every catalyst is placed on both sides of the chemical equation. However, we explicitly create states where the Ssa1/Ydj1 chaperones are bound to the protein, as opposed to acting as a catalyst, as this allows us to follow the pool of chaperones bound to protein aggregates. Regarding the chemical equations involving the chaperone, we take into account that larger aggregates bind more chaperones and assume a 1:1 chaperone to protein ratio. Following experimental observations, Hsp104 disaggregase is given a background concentration that increases upon accumulation of chaperone-hexamer complexes. We choose to model Hsp104 as a catalyst that does not have multiple bound and unbound states for simplicity.

In order to create full cell cycles, events were included to simulate Start and cell division when specific conditions are met. Inactivation of the Whi5 inhibitor by Cln3 is what triggers Start in the model. We assume that Whi5 has a decaying concentration during G1 (*Schmoller et al., 2015*) and, arbitrarily, Start is executed when 75% of Whi5 molecules are inactivated. The remainder of the cell cycle is assumed to take a constant time for simplicity. At division nuclear Cln3 is set to zero and Whi5 to the same initial Whi5 concentration in the previous generation, thus replicating the result that as cells age, the number of Whi5 molecules observed at the beginning of G1 increases with the increase in cell size (*Neurohr et al., 2018*).

Parameter selection was completed by scanning the parameter space using deterministic simulations in COPASI so that the average replicative lifespan matched the average value observed in experiments with wild-type cells. We then perturbed the model parameters by 1, 10% and 25% to obtain various cell cycle mutants, as described in *Supplementary file 3*. The most biologically accurate perturbation for each mutant in deterministic simulations was used for all subsequent simulations in stochastic mode. We also ensured that the parameters produced biologically accurate values for Ssa1/Ydj1, Wh5 and Cln3 (10000, 1000, and 100 molecules per cell, respectively). In order to speed up simulations, overall folded protein was limited to $10^5$ molecules per cell by increasing degradation of the folded protein pool, which is the final product of the process and does not affect the results of the model. Once the most biologically accurate parameter set was selected, time

course simulations were run using a direct stochastic method that implemented the Gillespie algorithm to simulate aggregation as a stochastic process. The stochastic time-course simulations were run 75 times for each condition, in order to obtain the distribution and average of replicative lifespans. The final parameter set is listed in *Supplementary file 2*. Due to the existence of backup mechanisms for *CLN3* (basal expression of *CLN1,2*) or *YDJ1* (*SIS1*), genetic ablation of these genes was simulated by applying different degrees of reduction in the concentrations of the respective proteins (*Supplementary file 3*). Increases and decreases in parameters were selected by conducting parameter sensitivity analysis in deterministic mode at 1, 10, 25% intervals after which the most biologically accurate perturbation was selected for stochastic simulations in each scenario. Initial sizes of mutants are based upon experimental data (*Ferrezuelo et al., 2012*).

## Miscellaneous

DNA-content distributions were obtained by Fluorescence Activated Cell Sorting (*Gallego et al., 1997*), with slight modifications to identify aged mother cells with labeled cell wall (see above) in a Gallios Flow Cytometer. Protein extracts were analyzed by SDS-PAGE (*Gallego et al., 1997*) or under semi-denaturing conditions to preserve amyloid aggregates by SDD-AGE (*Bagriantsev et al., 2006*). Immunoblot analysis with αGFP was performed as described (*Gallego et al., 1997*).

## Statistical analysis

Sample size is always indicated in the figure legend and, unless stated otherwise, median and quartile (Q) values are shown in all plots with single-cell data. Pairwise comparisons were performed with a Mann-Whitney U test; and the resulting p values are shown in the corresponding figure panels. For percentages, 95% confidence limits (CL) are always shown. Time-lapse data recorded from single cells in the CLiC microfluidics chamber are represented as the mean value of the population along time (with cells aligned at last budding event), while the shadowed area represent the 95% confidence limits of the mean. Logistic regression analysis was performed with the aid of a Java applet developed by J.C. Pezzullo (statpages.info/logistic.html).

## Data and software availability

The model was deposited in the BioModels (*Chelliah et al., 2015*) database as MODEL1901210001 in SBML format and a COPASI (*Hoops et al., 2006*) file to reproduce simulations with the parameter set shown in *Supplementary file 2*. BudJ (*Ferrezuelo et al., 2012*), CoinRICSJ (*Moreno and Aldea, 2019*) and the microcolony_size macro can be obtained as ImageJ (Wayne Rasband, NIH) plugins from ibmb.csic.es/groups/spatial-control-of-cell-cycle-entry. Jay Unruh's plugins can be obtained from research.stowers.org/imagejplugins. Prion propensity plots were obtained by the PAPA software (*Toombs et al., 2012*) located at combi.cs.colostate.edu/supplements.

## Acknowledgements

We thank E Rebollo, J Comas and M Kerexeta for technical assistance, and DE Gottschling, J Skotheim and KA Morano for providing strains. We also thank C Rose for editing the manuscript, and F Antequera, Y Barral, and C Gallego for helpful comments. This work was funded by the Ministry of Economy and Competitiveness of Spain, Consolider-Ingenio 2010, and the European Union (FEDER) to MA. KJ was supported by the EPSRC Centre for Doctoral Training in Cross-Disciplinary Approaches to Non-Equilibrium Systems (CANES, EP/L015854/1). DFM received an FI fellow of *Generalitat de Catalunya*.

## Additional information

### Funding

| Funder | Grant reference number | Author |
|---|---|---|
| Ministerio de Economía y Competitividad | BFU2016-80234-R | Martí Aldea |

The funders had no role in study design, data collection and interpretation, or the decision to submit the work for publication.

## Author contributions

David F Moreno, Conceptualization, Investigation, Methodology, Writing—original draft, Writing—review and editing; Kirsten Jenkins, Software, Investigation, Methodology, Writing—review and editing; Sandrine Morlot, Methodology, Writing—review and editing; Gilles Charvin, Supervision, Methodology, Writing—review and editing; Attila Csikasz-Nagy, Conceptualization, Formal analysis, Supervision, Investigation, Writing—original draft, Writing—review and editing; Martí Aldea, Conceptualization, Supervision, Funding acquisition, Investigation, Writing—original draft, Writing—review and editing

## Author ORCIDs

Sandrine Morlot http://orcid.org/0000-0002-0326-6694
Gilles Charvin http://orcid.org/0000-0002-6852-6952
Attila Csikasz-Nagy https://orcid.org/0000-0002-2919-5601
Martí Aldea https://orcid.org/0000-0002-8710-5336

## Decision letter and Author response

Decision letter https://doi.org/10.7554/eLife.48240.033
Author response https://doi.org/10.7554/eLife.48240.034

## Additional files

### Supplementary files

• Supplementary file 1. Chemical reactions of the integrative mathematical model.
DOI: https://doi.org/10.7554/eLife.48240.028

• Supplementary file 2. Parameter set of the integrative mathematical model.
DOI: https://doi.org/10.7554/eLife.48240.029

• Supplementary file 3. Parameter modifications to simulate different genotypes or relevant physiological conditions.
DOI: https://doi.org/10.7554/eLife.48240.030

• Transparent reporting form
DOI: https://doi.org/10.7554/eLife.48240.031

### Data availability

All data generated or analyzed during this study are included in the manuscript and supporting files. Source data files have been provided for all figures.

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
