## [Decision Letter]

Thank you for submitting your article "Proteostasis collapse halts G1 progression and delimits replicative lifespan" for consideration by *eLife*. Your article has been reviewed by three peer reviewers, including Yves Barral as the Reviewing Editor and Reviewer #1, and the evaluation has been overseen by David Ron as the Senior Editor. The following individual involved in review of your submission has agreed to reveal their identity: Thomas Nyström (Reviewer #2).

The reviewers have discussed the reviews with one another and the Reviewing Editor has drafted this decision to help you prepare a revised submission.

Summary

In this study, Moreno et al. establish that old yeast mother cells show a strong increase in proteostasis defects in their last divisions, as measured by chaperone mobility, levels and localisation. In parallel, these cells fail to properly fold the G1 cyclin Cln3, as reported by its failure to enter the nucleus, and the majority of the cells arrest their last cell cycle in the G1 phase of the cell cycle before dying. Cells induced to accumulate an aggregating protein show a similar increase in proteostasis defects, defect in Cln3 localisation to the nucleus and cell cycle arrest in G1. Based on a quantitative in silico model of this process, model that is is largely corroborated by the data, the authors propose that the collapse of proteostasis titrates the chaperones that normally ensure the proper activity and localisation of Cln3 and thereby lead to cell cycle arrest. Supporting this model, CLN3 over-expression extends the life-span of these cells. All three reviewers found that this manuscript is very well written, the experimental design is generally appropriate for testing the authors' hypotheses, and the experiments are well executed and convincing.

Essential revisions

The fact that CLN3 over-expression extends the longevity of the cells whereas overexpression of the chaperones does not is actually confusing. This seems at odds with the statement that chaperones are the limiting factor – does overproduction of Cln3 bypass the need for chaperones or what is going on? Does CLN3 over-expression promote expression of chaperones? If this is not the case, then the model needs to be amended at this time as it might not be as straightforward, and linear, a relationship as the authors suggest. Indeed, this observation could support two independent conclusions: Either proteostasis defects are accumulating more rapidly than any chaperone increase could control (and then it would be interesting to understand why), or one or several other processes limit the lifespan of the cell together with proteostasis and are not mitigated by chaperone over-expression.

Depending on the results of the additional experiments requested, the title of the paper might need to be amended.

[Editors' note: further revisions were requested prior to acceptance, as described below.]

Thank you for resubmitting your work entitled "Proteostasis collapse halts G1 progression and delimits replicative lifespan" for further consideration at *eLife*. Your revised article has been favorably evaluated by David Ron (Senior Editor) and three reviewers, one of whom is a member of our Board of Reviewing Editors.

The manuscript has been improved but there are some remaining issues that need to be addressed before acceptance, as outlined below:

While there is clear enthusiasm about your study and its findings, the reviewers were still unanimously concerned about whether the overall conclusion, as claimed in the title is truly supported by the data. The fact that Cln3 over-expression does prolongate the lifespan of wild type cells is a very interesting and important finding. However, the fact that chaperone over-expression does not show such effect remains a strong concern, and this is for two reasons.

1) It raises the question of whether the proteostasis collapse is THE process that delimits the replicative lifespan of yeast cells, i.e., is the ultimate cause of the death of the cells. Indeed, proteostasis defects could also be a symptom of another defect, which could be the actual lethal event. You very clearly show that proteostasis collapse can be lethal, as many other events, but you also clearly show that in that case chaperon over-expression does restore longevity. This strongly undermines the arguments made on the basis of in silico modelling.

2) It raises issues about the interpretation of how Cln3 over-expression promotes longevity. Indeed, if proteostasis collapse is the lethal event, how would cell cycle progression allow cells to escape it, in the context of your model of an exponential accumulation of unfolded proteins?

Therefore, the reviewers decided to insist on their request that you investigate whether CLN3 over-expression promotes chaperon expression. If this were not to be the case, the reviewers would find it more appropriate to discuss the issue more thoroughly and tone down the main conclusion of the paper.

---

## [Author Response]

Essential revisionsThe fact that CLN3 over-expression extends the longevity of the cells whereas overexpression of the chaperones does not is actually confusing. This seems at odds with the statement that chaperones are the limiting factor – does overproduction of Cln3 bypass the need for chaperones or what is going on? Does CLN3 over-expression promote expression of chaperones? If this is not the case, then the model needs to be amended at this time as it might not be as straightforward, and linear, a relationship as the authors suggest. Indeed, this observation could support two independent conclusions: Either proteostasis defects are accumulating more rapidly than any chaperone increase could control (and then it would be interesting to understand why), or one or several other processes limit the lifespan of the cell together with proteostasis and are not mitigated by chaperone over-expression.

As also suggested by the reviewers in their last minor point, we have added to main Figure 5 our data and model simulations corresponding to chaperone overexpression in aging cells. We have modified the text (subsection “Cln3 overexpression increases replicative lifespan in a chaperone-dependent manner” and Discussion paragraph five) to describe these data as follows: While Ydj1-deficient cells displayed a reduced lifespan as previously observed (Hill et al., 2014), concurrent overexpression of Ssa1 and Ydj1 did not increase lifespan significantly (Figure 5D). Albeit surprisingly, our model predicted that chaperone overexpression would have a very limited effect on lifespan (Figure 5D inset). By analyzing in detail the kinetics of Ydj1 levels and the appearance of protein aggregates in stochastic simulations (Figure 5E) we observed that, due the positive feedback loop inherent to autocatalytic aggregation, once the first protein aggregates appear they rapidly overcome Ydj1 levels by several orders of magnitude, thus making ineffective the relatively small (ca. 50%) increase in Ydj1 levels attained by GAL1p-driven overexpression (Yahya et al., 2014).

In addition, in the Discussion section (paragraph five) we mention that, similarly to our results with Hsp70/40, overexpression of Hsp104 restores proteasome activity in aging cells (Andersson et al., 2013) and suppresses lifespan defects of sir2 mutants (Erjavec et al., 2007), but does not increase lifespan significantly in wild-type cells (Andersson et al., 2013).

Finally, as we discuss in Discussion paragraph two, the accumulation of ERCs in aged cells (Shcheprova et al., 2008; Sinclair and Guarente, 1997) may play a more direct role repressing CLN1,2 expression (Neurohr et al., 2018). This would explain why, regarding lifespan extension, CLN3 overexpression is much more efficient than increasing chaperone expression levels.

Depending on the results of the additional experiments requested, the title of the paper might need to be amended.

We believe that, considering also our data above, the title reflects the main findings of our work, particularly in view of the important effects caused by precocious PD-protein aggregation in lifespan, and their suppression by either G1-cyclin or chaperone overexpression.

[Editors' note: further revisions were requested prior to acceptance, as described below.]

While there is clear enthusiasm about your study and its findings, the reviewers were still unanimously concerned about whether the overall conclusion, as claimed in the title is truly supported by the data. The fact that Cln3 over-expression does prolongate the lifespan of wild type cells is a very interesting and important finding. However, the fact that chaperone over-expression does not show such effect remains a strong concern, and this for two reasons.1) It raises the question of whether the proteostasis collapse is THE process that delimits the replicative lifespan of yeast cells, i.e., is the ultimate cause of the death of the cells. Indeed, proteostasis defects could also be a symptom of another defect, which could be the actual lethal event. You very clearly show that proteostasis collapse can be lethal, as many other events, but you also clearly show that in that case chaperon over-expression does restore longevity. This strongly undermines the arguments made on the basis of in silico modelling.

We agree with your first point that, although we find very strong temporal correlations, our data do not demonstrate that proteostasis collapse is THE reason why cell proliferation declines and cells arrest in G1 before death. We also accept that the initially proposed title was a bit ambiguous in this respect and we propose a new title as follows: “Proteostasis collapse, a hallmark of aging, hinders the chaperone-Start network and arrests cells in G1”. In line with this, the end of the Abstract has been modified to clarify the idea of a “molecular pathway postulating proteostasis decay as a key contributing effector of cell senescence.” In addition, although we had already addressed this point in our paper, the Discussion section has also been modified to emphasize the existence of other mechanisms as follows: “[…], the inability of chaperone overexpression to extend lifespan would also underscore the existence of chaperone-independent mechanisms affecting execution of Start in aged cells as above mentioned, such as ERC-mediated downregulation of G1/S genes (Neurohr et al., 2018).”

2) It raises issues about the interpretation of how Cln3 over-expression promotes longevity. Indeed, if proteostasis collapse is the lethal event, how would cell cycle progression allow cells to escape it, in the context of your model of an exponential accumulation of unfolded proteins?Therefore, the reviewers decided to insist on their request that you investigate whether CLN3 over-expression promotes chaperon expression. If this were not to be the case, the reviewers would find it more appropriate to discuss the issue more thoroughly and tone down the main conclusion of the paper.

Regarding your second point, we would like to point out that *CLN3* overexpression fully suppresses the requirement of the Ydj1 chaperone for cell cycle entry in young cells, with no effects on the expression of related chaperones (Vergés et al., 2007). We had interpreted this result as if retention of Cln3 at the ER was not a very efficient mechanism and could not cope with very high levels of translated Cln3. In any event, the above suppression would explain why, even if proteostasis is seriously affecting chaperone availability in very old cells, high levels of Cln3 would suffice to prolong replicative lifespan. We have added this comment to the Discussion section (paragraph five).